# Training Bayesian Neural Networks with Sparse Subspace Variational Inference

**Junbo Li[1], Zichen Miao[2], Qiang Qiu[2], Ruqi Zhang[1]**
1. Department of Computer Science 2. School of Electrical and Computer Engineering
Purdue University, West Lafayette, IN 47907, USA
`{ljunbo,miaoz,qqiu,ruqiz}@purdue.edu`

## Abstract

Bayesian neural networks (BNNs) offer uncertainty quantification but come with the downside of substantially increased training and inference costs. Sparse BNNs have been investigated for efficient inference, typically by either slowly introducing sparsity throughout the training or by post-training compression of dense BNNs. The dilemma of how to cut down massive training costs remains, particularly given the requirement to learn about the uncertainty. To solve this challenge, we introduce Sparse Subspace Variational Inference (SSVI), the first fully sparse BNN framework that maintains a consistently highly sparse Bayesian model throughout the training and inference phases. Starting from a randomly initialized low-dimensional sparse subspace, our approach alternately optimizes the sparse subspace basis selection and its associated parameters. While basis selection is characterized as a non-differentiable problem, we approximate the optimal solution with a removal-and-addition strategy, guided by novel criteria based on weight distribution statistics. Our extensive experiments show that SSVI sets new benchmarks in crafting sparse BNNs, achieving, for instance, a 10-20× compression in model size with under 3% performance drop, and up to 20× FLOPs reduction during training compared with dense VI training. Remarkably, SSVI also demonstrates enhanced robustness to hyperparameters, reducing the need for intricate tuning in VI and occasionally even surpassing VI-trained dense BNNs on both accuracy and uncertainty metrics.

## 1 Introduction

Bayesian neural networks (BNNs) (MacKay, 1992) infuse deep models with probabilistic methods through stochastic variational inference (Hoffman et al., 2013; Kingma et al., 2015; Molchanov et al., 2017; Dusenberry et al., 2020) or Monte-Carlo-based methods (Welling & Teh, 2011; Chen et al., 2014; Zhang et al., 2020e; Cobb & Jalaian, 2021; Wenzel et al., 2020; Zhao et al., 2019), facilitating a systematic approach to quantify uncertainties. Such a capacity to estimate uncertainty becomes important in applications where reliable decision-making is crucial (Filos et al., 2019; Feng et al., 2018; Abdullah et al., 2022). Unlike traditional neural networks, which offer point estimates, BNNs place distributions on weights and use a full posterior distribution to capture model uncertainty and diverse data representation. This has been shown to greatly improve generalization accuracy and uncertainty estimation across a wide variety of deep learning tasks (Zhang et al., 2020e; Blundell et al., 2015; Wilson & Izmailov, 2020; Vadera et al., 2022a).

However, shifting from deterministic models to BNNs brings its own set of complexities. The computational demands and memory requisites of BNNs, mainly stemming from sampling and integration over these weight distributions, are notably increased especially for large-scale neural networks (Zhang et al., 2020c;d;b;a). In standard deterministic neural networks, it has been observed that neural architectures can often be significantly condensed without greatly compromising performance (Frankle & Carbin, 2018; Evci et al., 2020). Drawing inspiration from this, some BNN approaches have leveraged sparsity-promoting priors, such as log-uniform (Kingma et al., 2015; Molchanov et al., 2017) or spike-and-slab (Deng et al., 2019; Wang et al., 2021), to gradually sparsify during training, or implement pruning after obtaining standard dense BNNs based on signal-to-noise ratios (Graves, 2011; Blundell et al., 2015; Ritter et al., 2021).

As a result, while they do mitigate the computational burden during inference, the substantial costs associated with training remain largely unaffected. Additionally, selecting optimal substructures (Vadera et al., 2022b) may be challenging using the one-shot pruning methods. In order to scale up BNNs to modern large-scale neural networks, a crucial question remains to be addressed:

*How can we craft a fully sparse framework for both training and inference in BNNs?*

In our pursuit of a solution, we depart from the conventional approach of designing complex sparse-promoting priors, instead, we advocate for a strategy that embeds sparsity into the posterior from the very beginning of training.

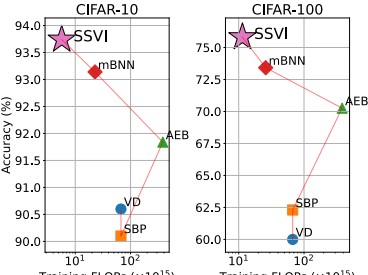

To this end, we introduce *Sparse Subspace Variational Inference* (SSVI) framework, where we confine the parameters of the approximate posterior to an adaptive subspace, performing VI solely within this domain, which significantly reduces training overheads. A defining feature of SSVI is its learning strategy, where both the parameters of the posterior and the subspace are co-learned, employing an alternating optimization technique. When updating the subspace, our method utilizes novel criteria that are constructed using the weight distribution statistics. Empirical results show that SSVI establishes superior standards in efficiency, accuracy, and uncertainty measures compared to existing methods, as shown in Figure 1. Moreover, SSVI displays robustness against hyperparameter variations and consistently showcases enhanced stability during training, sometimes even outperforming VI-trained dense BNNs. We summarize our contributions as the following:

Figure 1: Flops v.s. Accuracy. Our SSVI achieves the best test accuracy with the lowest training FLOPs.

- We propose SSVI, the first fully sparse BNN paradigm, to reduce both training and testing costs while achieving comparable and even better performance compared to dense BNNs.

- We introduce novel criteria for evaluating the importance of weights in BNNs, leveraging this information to simultaneously optimize the sparse subspace and the posterior.

- We set new benchmarks in performance metrics of BNNs, including efficiency, accuracy, and uncertainty estimation, achieving a 10-20× compression in model size with under 3% performance drop, and up to 20× FLOPs reduction during training. Through exhaustive ablation studies, we further refine and highlight the optimal design choices of our SSVI framework. We release the code at https://github.com/ljb121002/SSVI.

## 1.1 RELATED WORK

**Sparse BNNs** Since the introduction of the Bayesian learning framework, extensive research has focused on deriving efficient posterior inference methods (Tipping, 2001; Graves, 2011; Zhang et al., 2020c; Svensson et al., 2016). The challenge amplifies when transitioning to the domain of Bayesian neural networks (BNNs) (MacKay, 1992). To reduce computational costs, many studies have employed sparse-promoting priors to obtain sparse BNNs. Built upon variational inference, Molchanov et al. (2017) integrates variational dropout (Kingma et al., 2015) with a log-uniform prior, while Neklyudov et al. (2017) expands it to node-sparsity. Louizos et al. (2017), Ghosh et al. (2018) and Bai et al. (2020) further explore the use of half-Cauchy, Horseshoe and Gaussian-Dirac spike-and-slab priors respectively. Based on Markov chain Monte Carlo (MCMC), Deng et al. (2019); Wang et al. (2021) explore the use of a Laplace-Gaussian spike-and-slab prior. However, all these methods perform incremental pruning during training and rely heavily on intricately designed priors. Another line of work focuses on pruning fully trained dense BNNs (Graves, 2011; Blundell et al., 2015), thus the training costs still remain. The methodology most akin to ours is the recent work on masked BNN (mBNN) (Kong et al., 2023), which uses a similar birth-and-death mechanism for weight selection. However, mBNN requires a complex hierarchical sparse prior and uses MCMC to do posterior inference.

**Fully Sparse DNNs** While current sparse BNNs mainly rely on post-training or during-training pruning strategies due to the BNN training challenges, it is common in standard neural networks to engage in fully sparse training, essentially initiating with a sparse model. For instance, RigL (Evci

et al., 2020) starts with a sparse network, maintaining the desired sparsity throughout its training process by adjusting weights via dropping and growing. The criteria to determine such adjustments can derive from various factors, such as weight magnitude (Frankle & Carbin, 2018; Evci et al., 2020), gradient measures (Ström, 2015; Evci et al., 2020), and Hessian-based criteria (Hassibi & Stork, 1992; Singh & Alistarh, 2020).

## 2 PRELIMINARY

**Bayesian neural networks and variational inference**   We use the classical Bayesian neural network (BNN) formulation. Given a dataset $\mathcal{X} = (X, Y)$, where $X, Y$ are the independent and dependent variables respectively, we want to return the probability $p(\mathbf{x}|\mathcal{X})$, where $\mathbf{x} = (x, y)$ is a new data point. Let $\theta \in \mathbb{R}^d$ be the random variables of a Bayesian neural network, we have

$$p(\mathbf{x}|\mathcal{X}) = \int p(\mathbf{x}|\mathcal{X}, \theta) p(\theta|\mathcal{X}) d\theta \approx \frac{1}{N} \sum_{i=1}^{N} p(\mathbf{x}|\mathcal{X}, \theta^{(i)}), \quad \theta^{(i)} \sim p(\theta|\mathcal{X})$$

where $p(\theta|\mathcal{X})$ is the posterior distribution. In this paper, we use variational inference (VI) (Jordan et al., 1999) to approximate the true posterior. Specifically, we find a distribution $q_\phi(\theta)$, where $\phi$ are the distribution parameters, that minimizes $\text{KL}(q_\phi(\theta)||p(\theta|\mathcal{X}))$. VI typically adopts mean-field assumptions for both approximate posterior $q_\phi$ and prior $p$, which implies that $q_\phi(\theta) = \prod_{i=1}^{d} q_\phi(\theta_i)$ and $p(\theta) = \prod_{i=1}^{d} p(\theta_i)$. Thus, it is equivalent to minimizing:

$$\text{KL}(q_\phi(\theta)||p(\theta)) - \mathbb{E}_{q_\phi} \log p(\mathcal{X}|\theta) = \sum_{i=1}^{d} \text{KL}(q_\phi(\theta_i)||p(\theta_i)) - \mathbb{E}_{q_\phi} \log p(\mathcal{X}|\theta). \quad (1)$$

The expectation term is intractable and is usually approximated using Monte Carlo (MC) samples. In the backward process, the reparameterization trick (Kingma & Welling, 2013) is applied to enable gradient backpropagation for $\phi$. We use these standard assumptions and techniques in our method.

## 3 SPARSE SUBSPACE VARIATIONAL INFERENCE

In this section, we present our framework, sparse subspace variational inference (SSVI). First, we introduce the formulation and introduce an overview of the optimization in Section 3.1. Then we present the details of the alternative optimization strategy in Section 3.3.1 and 3.3.2.

### 3.1 SSVI FORMULATION

In conventional VI settings, the parameter vector $\phi$ generally has a dimensionality that is at least double that of $\theta$, accounting for both the mean and the variance. Consequently, there is a strong desire to seek sparse solutions to reduce the complexities arising from the augmented parameter space (Tipping, 2001). This holds particular significance when dealing with deep neural networks, where the model complexity is more pronounced (Molchanov et al., 2017; Neklyudov et al., 2017).

To address this, we introduce SSVI, wherein we confine the support of $q_\phi(\theta)$ to a subspace $S$ of a pre-assigned dimension $s$ within $\mathbb{R}^d$, implying $q_\phi(\theta) = 0$ for $\theta$ not belonging to $S$. To achieve a sparse model, we consider $S$ to be a subspace expanded directly from a subset of the axes of $\mathbb{R}^d$. We formulate SSVI as the following optimization problem

$$\min_{\phi, I} \quad KL(q_\phi(\theta)||p(\theta)) - \mathbb{E}_{q_\phi} \log p(\mathcal{X}|\theta) \quad (2)$$

$$\text{s.t.} \quad \forall i \notin I = \{n_1, \ldots, n_s\}, \quad q_\phi(\theta_i) = \delta(\theta_i).$$

In this formulation, $S$ is extended by a subset of the axes of $\mathbb{R}^d$ through the basis vectors $\{\mathbf{e}_{n_i}\}_{i=1}^{s}$. Here, $q_\phi(\theta_j)$ refers to the marginal density function of $\theta_j$ according to $q_\phi(\theta)$, and $\delta(\cdot)$ stands for the Dirac delta function. This means for $i \notin \{n_1, \cdots, n_S\}$, the approximate posterior distribution of $\theta_i$ shrinks to zero. For simplicity, we consider a Gaussian distribution for $q_\phi$ where $\phi \in \mathbb{R}^{d \times 2}$ with $\phi_i = (\mu_i, \sigma_i^2)$ representing the mean and variance for $1 \le i \le d$. Other mean-field variational distributions can also be combined with our method. This suggests that, for $i \notin \{n_1, \cdots, n_S\}$, $\phi_i =$

$(0, 0)$, resulting in a sparse solution. Equivalently, we can write (2) as the following optimization problem. Here $\odot$ means element-wise multiplication. The constraint conditions mean that when $\gamma_i$ is 0, $\mu_i$ and $\sigma_i$ are also 0.

$$\min_{\substack{\phi=(\mu, \sigma^2)\in\mathbb{R}^{d\times 2}, \\ \gamma\in\{0,1\}^d\subset\mathbb{R}^d}} \quad \sum_{i=1}^{d} \gamma_i KL(q_\phi(\theta_i)\|p(\theta_i)) - \mathbb{E}_{q_\phi}\log p(\mathcal{X}|\theta) \tag{3}$$

$$\text{s.t.} \quad \|\gamma\|_1 = s, \quad \mu = \mu \odot \gamma, \quad \sigma^2 = \sigma^2 \odot \gamma$$

The formulation (3) has three key advantages compared to previous sparse BNN methods (Kingma et al., 2015; Molchanov et al., 2017; Neklyudov et al., 2017; Kong et al., 2023), which we discuss in detail separately below.

**Reduce training costs with constant sparsity rate** Previous methods in both VI and MCMC-based sparse BNNs typically start with a dense model and traverse the full parameter space to eventually obtain sparse solutions — a process finalized through post-training truncations based on specific criteria. In contrast, our SSVI approach optimizes parameters directly within a sparse subspace. This ensures a constant sparsity level of $s/d$ throughout training, thereby considerably diminishing both computational and memory costs during training.

**Guarantee to achieve target sparsity levels** Existing approaches in both VI and MCMC-based sparse BNNs depend on unpredictable sparsity regularization from the prior and post-hoc truncation based on threshold values. This leads to indeterminable sparsity levels, often requiring extensive tuning of the prior to achieve desired levels of sparsity. In contrast, our method provides a straightforward means of controlling sparsity through the hyperparameter $s$. By constraining the search space to only subspaces with a dimension of $s$, any solution pair $(\phi, \gamma)$ of (3) guarantees the desired level of sparsity.

**Allow the use of Gaussian distribution for both prior and variational posterior** Our formulation does not require an intricated design for either the prior or the posterior. Instead, we can simply adopt commonly used variational distributions, such as Gaussian distributions, for both the prior and posterior distributions. Furthermore, this simplifies the computation by enabling the direct calculation of the KL divergence term without necessitating complicated and expensive approximations, which are often required when using sparse-promoting priors.

### 3.1.1 ALTERNATIVE OPTIMIZATION PROCEDURE

The optimization problem in (3) includes both $\phi$ and $\gamma$. Denote $I = \{i : \gamma_i = 1\}$. At the beginning, we initialize $\gamma$ by randomly selecting a subset $I^0 = \{n_1, \ldots, n_s\}$ from $\{1, \ldots, d\}$ where indices $\gamma_i$ are set to 1. We also initialize $\phi^0$ on $I^0$ following techniques commonly used in VI-based BNN training. Given the discrete nature of $\gamma$, directly applying gradient descent for both variables is infeasible. To address this challenge, we propose to alternatively update $\phi$ and $\gamma$ at each iteration. The optimization for $\phi$ can be done via standard VI-based BNN training, leveraging gradient information. However, updating $\gamma$ represents a more sophisticated task. We first present our algorithm in Algorithm 1, and we will delve into the specifics of the design of the updating processes in the forthcoming sections.

### 3.2 UPDATE THE VARIATIONAL PARAMETER $\phi$

Given a fixed $\gamma$, the update for $\phi$ is simple and can be achieved using optimization methods like SGD. We obtain $\phi^{t+1}$ from $(\phi^t, \gamma^t)$ through $M$ gradient descent steps, and each step is similar to the standard VI update.

During the forward process, we estimate $\mathbb{E}_{q_\phi}\log p(\mathcal{X}|\theta)$ using Monte Carlo samples from $q_\phi(\theta)$. For improved accuracy, distinct $\theta$ samples are drawn for each input in a batch. However, naive methods, which require a forward pass per sample, are inefficient. We thus employ the local reparameterization trick (LRT) (Kingma et al., 2015; Molchanov et al., 2017) to improve efficiency, achieving equivalent outputs in just two forward processes. Specifically, for a linear operation with inputs $x \in \mathbb{R}^{p\times B}$ and Gaussian matrix $A \in \mathbb{R}^{p\times q}$, where $B$ is batch size, $p, q$ are the input and

---

**Algorithm 1** Sparse Subspace Variational Inference (SSVI)

---

**Require:** A BNN $\theta \in \mathbb{R}^d$ with prior $p(\theta)$, variational distribution $q_\phi(\theta)$, target sparsity $s/d$, replacement rate $\{r_t\}$, inner update steps $M$, total steps $T$.
1: Random initialize $(\phi^0, \gamma^0)$ from the feasible set of (3), and set $\gamma^{-1} = \gamma^0$.
2: **for** $t = 0, \ldots, T$ **do**
3:     # Update $\phi$.
4:     $\phi^{t,0} = \texttt{Initialize}(t, \phi^t, \gamma^t, \gamma^{t-1})$ according to Section 3.2.
5:     **for** $m = 0, \ldots, M - 1$ **do**
6:         Obtain $\phi^{t,m+1}$ using the gradient of (3).
7:     **end for**
8:     $\phi^{t+1} = \phi^{t,M}$.
9:     # Update $\gamma$.
10:    $\gamma^t_{\text{remove}} = \texttt{Removal}(\gamma^t, \phi^{t+1}, r_t)$ according to Section 3.3.1.
11:    $\gamma^{t+1} = \texttt{Addition}(\gamma^t_{\text{remove}}, \phi^{t+1}, r_t)$ according to Section 3.3.2.
12: **end for**

---

output dimensions, $\mu$ and $\sigma$ define $A$'s mean and variance, the output $y$ is computed by

$$y = \mu \cdot x + \sqrt{(\sigma \odot \sigma) \cdot (x \odot x)} \odot \varepsilon, \tag{4}$$

where $\varepsilon$ is sampled from standard Gaussian noise matching $y$'s dimension. This method mirrors sampling distinct weights for each input $x_{:,b}$ via just two forwards, rather than $B$ times. While the forward process matches the naive approach, LRT's backward process for $\sigma$ differs and influences the SSVI design. Specifically, the gradient for $\sigma$ with respect to the received loss $l$ becomes:

$$\frac{\partial l}{\partial \sigma} = \frac{\partial l}{\partial y} \cdot \left( \frac{\sigma \cdot (x \odot x)}{\sqrt{(\sigma \odot \sigma) \cdot (x \odot x)}} \odot \varepsilon \right). \tag{5}$$

Please refer to Appendix A for a detailed discussion.

**Establishing initial values for $\phi^{t+1}$ updates** We find that (5) exhibits a linear dependency on $\sigma$. In the context of SSVI, the sparsity ratio of $\phi$ remains constant throughout the training process. Consequently, for all $t \geq 1$ and $j$ satisfying $\gamma^t_j - \gamma^{t-1}_j = 1$ (referring to the indices freshly updated to be non-zero), the value of $\sigma^t_j$ is zero. This implies that the straightforward application of gradient descent is infeasible as the gradient persistently equals zero for $\sigma^t_j$. Therefore, it is crucial to design a strategy to set the initial value for $\sigma^t_j$, different from the conventional sparse training in standard DNNs which does not require such initial values.

We consider two potential strategies. The initial approach sets $\sigma^t_j$ to a very small number close to zero, which helps stabilize training while avoiding the zero gradient issue. Alternatively, $\sigma^t_j$ can be set to the average value of the non-zero $\sigma_k$, derived from the same module — that is, the identical convolutional kernel or the same fully connected layer as $\sigma^t_j$. Favorably, the latter strategy does not demand additional hyperparameters and has demonstrated superior stability in experimental settings, thereby being our method's default choice.

### 3.3 UPDATE THE SPARSE SUBSPACE $\gamma$

The set of non-zero elements in $\gamma$ outlines the architecture of the subnetwork. Since $\gamma$ takes discrete values, it cannot be updated using regular gradient-based methods. Based on the existing $(\phi^{t+1}, \gamma^t)$, we propose to obtain $\gamma^{t+1}$ through a process of selective removal and addition of indices in $\gamma^t$. This adjustment process takes inspiration from traditional sparse training techniques seen in regular, non-Bayesian neural network training (Frankle & Carbin, 2018; Evci et al., 2020). Notably, we introduce several novel techniques that are especially compatible with BNN training.

#### 3.3.1 SUBSPACE REMOVAL

To derive $\gamma^{t+1}$, we begin by identifying and eliminating less significant indices in $\gamma^t$ based on $\phi^{t+1} = (\mu^{t+1}, \sigma^{t+1^2}) \in \mathbb{R}^{d \times 2}$. In standard DNN sparse training, one common approach is to prune

weights with small magnitudes, under the assumption that small weights tend to have a negligible impact on the output. Building upon this insight, we derive several novel criteria for removing indices in the subspace. We omit the step $t$ in the following.

**Criteria 1: $|\boldsymbol{\mu}|$** A straightforward adaptation for BNNs is to remove indices using the absolute values of the mean, *i.e.*, $|\mu|$, to set $\gamma_i$ with the smallest $|\mu_i|$ to be zero. However, this strategy overlooks the critical information about uncertainty encoded in $\sigma^2$.

**Criteria 2: $\text{SNR}_{q_\phi}(\boldsymbol{\theta}) = |\boldsymbol{\mu}|/\boldsymbol{\sigma}$** Another commonly used metric to decide the importance of weights in BNN is the Signal-to-Noise Ratio (SNR) (Kingma et al., 2015; Neklyudov et al., 2017), *i.e.*,$|\mu|/\sigma$. This metric considers both the magnitude and variance of the weights. Though previous works have utilized SNR for post-training pruning (Kingma et al., 2015; Neklyudov et al., 2017; Louizos et al., 2017) , to the best of our knowledge, its application during training is novel.

**Criteria 3: $\mathbb{E}_{q_\phi}|\boldsymbol{\theta}|$** We introduce a new criterion that considers both the mean and the uncertainty. Criteria 1 $|\mu|$ essentially equates to $|\mathbb{E}_{q_\phi}\theta|$. However, in BNN, where weights are randomly sampled before each forward pass with given inputs, it is more appropriate to use $\mathbb{E}_{q_\phi}|\theta|$. The reasoning behind this is that weights with smaller absolute values, on average, have lesser influence on the outputs. Notably, this criterion can be explicitly calculated when $q_\phi$ follows a Gaussian distribution. Specifically, for $i$-th dimension of weight $\theta_i$ with mean $\mu_i$ and variance $\sigma_i^2$, we have

$$\mathbb{E}_{q_\phi}|\theta_i| = \mu_i\left(2\Phi\left(\frac{\mu_i}{\sigma_i}\right) - 1\right) + \frac{2\sigma_i}{\sqrt{2\pi}}\exp\left(-\frac{\mu_i^2}{2\sigma_i^2}\right), \text{ where } \Phi(x) := \int_{-\infty}^{x}\frac{1}{\sqrt{2\pi}}\exp\left(-\frac{y^2}{2}\right)dy. \tag{6}$$

The full derivation can be found in Appendix B. This criterion differs significantly from Criteria 1 by expressing a theoretically-backed new rule that combines the magnitude insights from Criteria 1, represented by $|\mu|$, with the uncertainty dimensions from Criteria 2, denoted by $|\mu|/\sigma$. We highlight that the role of $\mu/\sigma$ is deliberately bounded in this formula, either by the cumulative distribution function $\Phi(\cdot)$ or by the inverse of the exponential function $1/\exp(\cdot)$. Given that the uncertainty measure $\sigma$ can be challenging to estimate accurately during initial training phases, this imposed boundary on $\mu/\sigma$ prevents it from excessively skewing weight prioritization, thus helping to address potential issues of model instability. Besides the constrained behavior of $\mu/\sigma$, the criteria prominently lean towards $\mu_i$ because of its linear relationship. However, a concerning aspect is the near-linear dependency on $\sigma_i$. This is problematic as a high $\sigma_i$ typically indicates a weight's insignificance. To counteract this, we propose subsequent criteria that introduce regularization.

**Criteria 4: $\text{SNR}_{q_\phi}(|\boldsymbol{\theta}|)$** Although $\mathbb{E}_{q_\phi}|\theta|$ already contains uncertainty information, we can further consider the SNR of $|\theta|$ as opposed to $\theta$ in Criteria 2. For $\theta_i$, we have

$$\text{Var}_{q_\phi}(|\theta_i|) = \mathbb{E}_{q_\phi}|\theta_i|^2 - \left(\mathbb{E}_{q_\phi}|\theta_i|\right)^2 = \mathbb{E}_{q_\phi}\theta_i^2 - \left(\mathbb{E}_{q_\phi}|\theta_i|\right)^2 = \sigma_i^2 + \mu_i^2 - \left(\mathbb{E}_{q_\phi}|\theta_i|\right)^2.$$

Hence, the SNR is

$$\text{SNR}_{q_\phi}(|\theta_i|) = \frac{\mu_i\left(2\Phi\left(\frac{\mu_i}{\sigma_i}\right) - 1\right) + \frac{2\sigma_i}{\sqrt{2\pi}}\exp\left(-\frac{\mu_i^2}{2\sigma_i^2}\right)}{\sqrt{\sigma_i^2 + \mu_i^2 - \left[\mu_i\left(2\Phi\left(\frac{\mu_i}{\sigma_i}\right) - 1\right) + \frac{2\sigma_i}{\sqrt{2\pi}}\exp\left(-\frac{\mu_i^2}{2\sigma_i^2}\right)\right]^2}}. \tag{7}$$

Given that the denominator of (7) progresses linearly with respect to $\sigma$, it counteracts the unfavorable characteristic seen in (6). Additionally, $\mu$ retains its prominent role as the denominator behaves sublinearly with $\mu$, whereas the numerator exhibits a linear relationship.

**Criteria 5 & 6: $\mathbb{E}_{q_\phi}e^{\lambda|\boldsymbol{\theta}|}$ and $\text{SNR}_{q_\phi}(e^{\lambda|\boldsymbol{\theta}|})$** When examining weight importance in the network, one could go beyond the straightforward metrics like the expectation and SNR of $|\theta|$ and delve into more general functions $f$. These functions should be monotonic within the range $[0, +\infty)$ to ensure that $\theta$ with greater absolute values always has high importance. As a practical example, we adopt the function $f(|\theta|) = \exp(\lambda|\theta|)$, where $\lambda$ is a positive hyperparameter. This choice results

Table 1: Comparisons with dense BNNs. Our method achieves 10-20x savings in both training costs and model complexity via keeping high-sparsity subspaces throughout the training. Here we show the results with two different sparsity levels as inputs of SSVI.

| Dataset | Model | Accuracy | ECE | Sparsity | Training FLOPs |
|---------|-------|----------|-----|----------|----------------|
| CIFAR-10 | Dense BNN | 95.13 | 0.004 | 0% | 1x($1.1 \times 10^{17}$) |
| | w/ SSVI | 94.58 | 0.005 | **90%** | **0.10x** |
| | w/ SSVI | 93.69 | 0.006 | **95%** | **0.05x** |
| CIFAR-100 | Dense BNN | 78.20 | 0.0016 | 0% | 1x($1.2 \times 10^{17}$) |
| | w/ SSVI | 75.81 | 0.0017 | **90%** | **0.10x** |
| | w/ SSVI | 73.83 | 0.0018 | **95%** | **0.05x** |

in an explicit calculation of both the expectation and SNR of $f(|\theta|)$ under a Gaussian distribution without necessitating further approximations. For $\lambda > 0$, the derived outcomes are outlined below:

$$\mathbb{E}_{q_\phi} e^{\lambda|\theta|} = \Phi\left(\frac{\mu}{\sigma} + \lambda\sigma\right) e^{\frac{\lambda^2\sigma^2}{2} + \lambda\mu} + \Phi\left(-\frac{\mu}{\sigma} + \lambda\sigma\right) e^{\frac{\lambda^2\sigma^2}{2} - \lambda\mu},$$

$$\text{SNR}_{q_\phi}(e^{\lambda|\theta|}) = \tag{8}$$

$$\frac{\Phi\left(\frac{\mu}{\sigma} + \lambda\sigma\right) e^{\frac{\lambda^2\sigma^2}{2} + \lambda\mu} + \Phi\left(-\frac{\mu}{\sigma} + \lambda\sigma\right) e^{\frac{\lambda^2\sigma^2}{2} - \lambda\mu}}{\sqrt{\Phi\left(\frac{\mu}{\sigma} + 2\lambda\sigma\right) e^{2\lambda^2\sigma^2 + 2\lambda\mu} + \Phi\left(-\frac{\mu}{\sigma} + 2\lambda\sigma\right) e^{2\lambda^2\sigma^2 - 2\lambda\mu} - \left(\Phi\left(\frac{\mu}{\sigma} + \lambda\sigma\right) e^{\frac{\lambda^2\sigma^2}{2} + \lambda\mu} + \Phi\left(-\frac{\mu}{\sigma} + \lambda\sigma\right) e^{\frac{\lambda^2\sigma^2}{2} - \lambda\mu}\right)^2}}.$$

**Criteria Selection** Criteria 4 $\text{SNR}(|\theta|)$ has a compelling theoretical explanation, and our empirical results in Section 4 further support its superior and stable performance during training compared to other criteria. Consequently, we choose it as the default for SSVI.

### 3.3.2 Subspace addition

After pruning spaces using one of the proposed criteria, it is essential to reintroduce some of the removed spaces back to $\gamma$. Drawing inspiration from traditional sparse training in DNNs, we select important weights by their gradient's absolute magnitude. Specifically, for BNN training with a stochastic batch, denoting the loss function in (3) as $f_{\theta,\gamma}(x)$, where $\theta$ is a sample for a stochastic batch $x$ with batch size $B$, the selection criterion is given by:

$$\mathbb{E}_{q_\phi}\left|\mathbb{E}_x \frac{1}{B}\sum_{i=1}^{B}\nabla_\theta f_{\theta,\gamma}(x_i)\right|. \tag{9}$$

To compute the two expectations, we propose three criteria, using Monte Carlo approximation:

1. Apply a one-step Monte-Carlo for $(\theta, x)$: $\left|\frac{1}{B}\sum_{i=1}^{B}\nabla_\theta f_{\theta,\gamma}(x_i)\right|$.

2. Apply a one-step Monte-Carlo for $x$ with $\theta$ set to $\mu$: $\left|\frac{1}{B}\sum_{i=1}^{B}\nabla_\theta f_{\mu,\gamma}(x_i)\right|$.

3. Apply a multi-step Monte-Carlo for $(\theta, x)$: $\frac{1}{K}\sum_{k=1}^{K}\left|\frac{1}{B}\sum_{i=1}^{B}\nabla_\theta f_{\theta^k,\gamma}(x_i^k)\right|$.

Ablation studies in Section 4 indicate that Criteria 1, while being the most straightforward, delivers performance on par with the other two. Thus, we default to Criteria 1 for SSVI.

## 4 Experiments

### 4.1 Main results

We conduct experiments using CIFAR-10 and CIFAR-100 datasets (Krizhevsky, 2009) and ResNet-18 (He et al., 2016) as the backbone networks. To facilitate stable training, we use KL warm-up (Molchanov et al., 2017; Sønderby et al., 2016), a technique widely applied in VI-based BNNs. Specifically, an auxiliary coefficient $\beta$ is introduced for the KL term in Eq.(1), gradually increasing

Table 2: Comparison with previous sparse BNNs on CIFAR-10 and CIFAR-100. SSVI achieves the best accuracy and uncertainty quantification with the lowest computation and memory costs.

| Dataset | Algorithm | Accuracy | Loss | ECE | Sparsity | Complexity | Training FLOPs ($\times 10^{15}$) |
|---------|-----------|----------|------|-----|----------|------------|------------------------------------|
| CIFAR-10 | VD (Louizos et al., 2017) | 90.6 | 0.298 | 0.009 | 76.8% | 2.60M | 67.1 |
| | SBP (Neklyudov et al., 2017) | 90.1 | 0.301 | 0.009 | 81.1% | 2.12M | 67.1 |
| | AEB (Deng et al., 2019) | 91.84 | 0.308 | 0.023 | 90.0% | 1.2M | 383.4 |
| | mBNN (Kong et al., 2023) | 93.14 | 0.227 | 0.008 | 96.4% | 0.41M | 22.8 |
| | **SSVI** | **93.74** | **0.218** | **0.006** | 95.0% | 0.56M | **5.7** |
| CIFAR-100 | VD (Louizos et al., 2017) | 60.0 | 1.977 | 0.0060 | 58.3% | 4.66M | 67.1 |
| | SBP (Neklyudov et al., 2017) | 62.3 | 1.834 | 0.0053 | 61.2% | 4.34M | 67.1 |
| | AEB (Deng et al., 2019) | 70.26 | 1.103 | 0.0110 | 90% | 1.2M | 383.4 |
| | mBNN (Kong et al., 2023) | 73.42 | 1.078 | 0.0024 | 87.8% | 1.37M | 25.9 |
| | **SSVI** | **75.81** | **0.983** | **0.0017** | 90% | **1.12M** | **11.4** |

from 0 to $\beta_{\max}$ during training. More training details are shown in Appendix C. For inference, we consider accuracy, loss, and expected calibration error (ECE) (Guo et al., 2017) as the metrics, which are obtained by five samples from the posterior distribution following prior work Kong et al. (2023). We also do evaluations on distribution shift benchmarks in Appendix D.1. For all the experiments, we train with three different seeds and average the results.

We first demonstrate the results of the reduction in training costs and model complexity compared with dense BNNs in Table 1. SSVI achieves comparable test accuracy and ECE with drastically reduced model size and training costs. For example, SSVI achieves 20x model size compression and FLOPs reduction with under 2% accuracy drop and 0.002 ECE increase on CIFAR-10, and 10x model size compression and FLOPs reduction with under 3% accuracy drop and 0.0001 ECE increase on CIFAR-100. We also compare our method with previous sparse BNN methods in Table 2. The results show that SSVI significantly outperforms all previous works across all metrics on both datasets, achieving the best accuracy and uncertainty quantification with the lowest computation and memory costs. It strongly indicates that our method, optimizing both the subspace and variational parameters, can lead to better sparse solutions than using complicated sparse-promoting priors.

## 4.2 ANALYSIS AND ABLATION STUDIES

In this section, we perform a thorough analysis of SSVI's properties and present ablation studies focusing on its core designs. Here we train ResNet-18 on the CIFAR-100 dataset, which is more challenging than CIFAR-10, and can help distinguish different algorithm designs.

**Flexible target sparsity**   Different from previous works (Molchanov et al., 2017; Kong et al., 2023), SSVI can flexibly assign the target sparsity before training. In Figure 2, we plot the results on a broad range of sparsity from 50% to 1%. The plots show that our SSVI is robust to different sparsity levels with minimal accuracy drop. For a clearer perspective, we contrast our results with RigL (Evci et al., 2020), a method closely aligned with our approach in the domain of sparse training for traditional DNNs. While both methods employ similar subspace training techniques, our criteria are tailored for BNNs. The results indicate that while the performance trajectory of our BNNs in relation to sparsity closely mirrors that of RigL, our approach is more robust to varying sparsity levels, as evidenced by a lesser decline in accuracy.

**SSVI's edge over VI**   While SSVI inherently optimizes parameters within a subspace, making it more efficient than standard VI, it even occasionally outperforms VI. This superior performance is largely due to its enhanced resilience to hyperparameter variations. To illustrate, consider VI with Gaussian distributions where the parameters are represented as $\phi = (\mu, \sigma)$. The optimization process requires an initialization $\phi^0 = (\mu^0, \sigma^0)$ where $\sigma^0$ is typically sampled from a Gaussian with minimal mean and variance. Ensuring small $\sigma^0$ is important for preventing training failures in the initial phase. Our results further verify the instability of BNN training with VI to initial values, highlighting the need for precise hyperparameter tuning. In contrast, BNNs trained with SSVI demonstrate increased tolerance to deviations in the initial configurations, thus simplifying the intricacies associated with hyperparameter optimization.

Our findings are presented in Figure 2. We use SSVI with sparsity 0.1 with different removal criteria. By adjusting the initialization mean of $\sigma^0$ from 0.001 to 0.01 while keeping all other settings unchanged for both VI and SSVI, we clearly see that as the mean marginally increases, VI's performance deteriorates immediately. In contrast, SSVI maintains robustness across all different dropping criteria and even outperforms VI at a significantly reduced training expense.

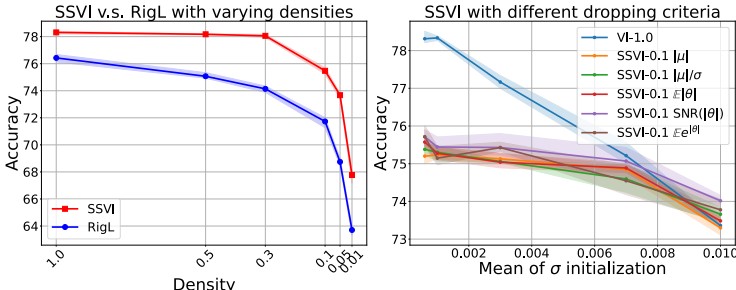

Figure 2: Left: the performance of SSVI for BNNs compared to RigL (Evci et al., 2020) for standard NNs across various sparsity levels. SSVI demonstrates superior robustness to different sparsity levels. Right: the results for VI at full density and SSVI at 0.1 density using different removal criteria. We see that VI is heavily dependent on precise hyperparameter tuning, especially concerning KL warmup, whereas SSVI offers more consistent training across all drop criteria without additional tricks, and outperforms VI at initialization value 0.01. Notably, among all criteria, $\mathrm{SNR}(|\theta|)$ stands out, leveraging uncertainty information in a theoretically backed manner.

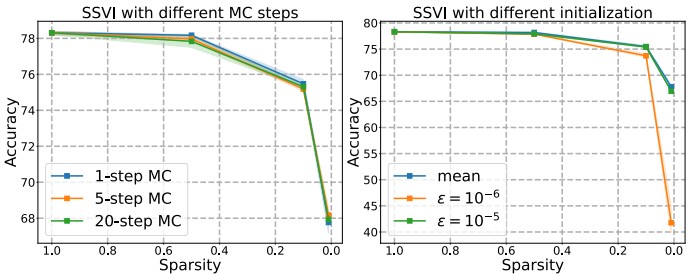

Figure 3: Left: ablation results concerning the number of MC steps, indicating minimal differences between them. Right: ablations related to initialization, highlighting the advantage of initializing with the mean.

**Ablation study on removal and addition criteria** In Section 3.3.1, we introduce several criteria tailored for BNNs using weight distribution statistics. Figure 2 compares accuracy for these criteria. We see that $\mathrm{SNR}_{q_\phi}(|\theta|)$ achieves the best accuracy on all sparsity levels, aligning with our analysis in Section 3.3.1. We also calculate the Intersection over Union (IoU) among the weights removed based on each criteria pair. See Appendix D.2 for details. Figure 3 shows the results using different MC steps, highlighting that multi-step MC is not better than the 1-step MC. Thus, we advocate for the one-step MC due to its superiority in both efficacy and performance.

**Ablation study on updating** $\sigma$ In Section 3.2, we highlighted the necessity for a specialized design in initializing the variance parameters $\sigma$ in $\phi$ before its update using gradient descent, as opposed to starting with zero, as driven by the gradient outlined in (5). Two strategies were proposed: one starts with a small value close to zero, and the other uses the average of the remaining non-zero $\sigma$ values. The comparison results of these strategies, while holding other factors the same, are showcased in Figure 3, the optimal hyperparameter setting for the first method appears to closely mirror the results when $\phi$ is initialized using the mean of non-zero $\sigma$ values. Therefore, we adopt the mean-based initialization approach, which also eliminates the necessity for extra hyperparameters.

## 5 CONCLUSION

In this paper, we propose Sparse Subspace Variational Inference (SSVI), the first fully sparse framework for both training and inference in Bayesian neural networks (BNNs). Specifically, SSVI optimizes the sparse subspace and variational parameters alternately, employing novel criteria based on weight distribution statistics for the removal and addition within the subspace. We conduct comprehensive experiments to demonstrate that SSVI establishes new benchmarks in sparse BNNs, by significantly outperforming previous methods in terms of both performance and efficiency. We also perform in-depth analysis to explore SSVI's properties and identify optimal design choices for criteria. As we look ahead, our future endeavors will focus on refining the optimization process. This could involve designing new weighting functions for the SNR removal criteria and exploring new addition criteria that utilize higher-order information of the weight distribution.

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
