## A   DETAILS OF LOCAL REPARAMETERIZATION TRICK

For the sake of completeness, we show the details of local reparameterization trick (LRT) here. Initially, we examine the forward process of SSVI and determine the distribution of the outputs. Through this, we introduce LRT, which leads to the same distribution with the naive implementation. Subsequently, we highlight the disparities in the backward processes between the naive approach and LRT, underscoring the importance of appropriate initialization.

**Forward process**   Similar to the conventional VI, the term $\mathbb{E}_{q_\phi} \log p(\mathcal{X}|\theta)$ must be approximated using Monte Carlo by drawing samples of $\theta$ from $q_\phi(\theta)$. While it's advantageous to draw distinct $\theta$ samples for various inputs within a batch, this approach complicates parallel computing operations. To resolve this challenge, we employ the local reparameterization trick following (Kingma et al., 2015; Molchanov et al., 2017). To elaborate, consider a single linear operation as an illustration. Given inputs $x \in \mathbb{R}^{p \times B}$, characterized by a feature dimension of $p$ and a batch size of $B$, alongside a randomly Gaussian distributed matrix $A \in \mathbb{R}^{p \times q}$ which is randomly sampled for each $1 \leq b \leq B$. Here, the matrices $\mu$ and $\sigma \in \mathbb{R}^{p \times q}$ delineate the mean and variance for each corresponding index of $A$. Consequently, the corresponding output $y$ also adheres to a Gaussian distribution, and for every sample $1 \leq b \leq B$ we have:

$$\mathbb{E}y_{:,b} = \mathbb{E}A \cdot x_{:,b} = \mu \cdot x_{:,b},$$
$$Var(y_{:,b}) = Var(Ax_{:,b}) = (\sigma \odot \sigma) \cdot (x_{:,b} \odot x_{:,b}). \tag{10}$$

Observe that $\mu$ and $\sigma \odot \sigma$ are both irrelevant with $b$, which means the calculations for both $\mathbb{E}y$ and $Var(y)$ can be carried out in parallel for all samples in the batch. This necessitates only two forward processes, one for the mean and another for the variance, as opposed to the original method which demands $B$ separate forward processes for each sample. Consequently, this approach diminishes computational demands remarkably.

**Backward process**   The sampling procedure is non-differentiable. Traditional approaches to circumvent this involve the reparameterization trick, where noise, denoted as $\varepsilon$, is sampled from a standard normal distribution, $\mathcal{N}(0, 1)$. The forward process is then represented as $\mu + \sigma\varepsilon$, facilitating the direct computation of gradients for both $\mu$ and $\sigma$. Concretely in our case, for each sample in the range $1 \leq b \leq B$, we generate a standard Gaussian matrix $\eta^{(b)} \in \mathbb{R}^{p \times q}$ and derive the respective output:

$$y_{:,b} = (\mu + \sigma \odot \eta^{(b)})x_{:,b} \tag{11}$$

Different from (11), our approach to the local reparameterization trick retains the core concept but diverges in its implementation. Initially, we calculate both the expected value $\mathbb{E}y$ and the variance $Var(y)$. Subsequently, we sample a standard Gaussian matrix $\varepsilon \in \mathbb{R}^{q \times b}$ and set:

$$y = \mu \cdot x + \sqrt{(\sigma \odot \sigma) \cdot (x \odot x)} \odot \varepsilon. \tag{12}$$

As previously noted, equations (11) and (12) adhere to the same distribution, with (12) offering substantial efficiency due to its parallel implementation. However, it is essential to highlight that, although the forward processes of the two are equivalent, they yield different gradients with respect to $\sigma$. We show the different gradient updates in the following and emphasize their significant role in influencing the updates of $\phi$ in SSVI. The gradient for $\mu$ is the same for both (11) and (12). For $\sigma$, the gradient in (11) is:

$$\frac{\partial l}{\partial \sigma} = \frac{\partial l}{\partial y} \cdot \left( \sum_{b=1}^{B} \eta^{(b)} \cdot x_{:,b} \right). \tag{13}$$

And the gradient in (12) is:

$$\frac{\partial l}{\partial \sigma} = \frac{\partial l}{\partial y} \cdot \left( \frac{\sigma \cdot (x \odot x)}{\sqrt{(\sigma \odot \sigma) \cdot (x \odot x)}} \odot \varepsilon \right). \tag{14}$$

## B   DERIVATION FOR DROPPING CRITERIA

In the following, we omit the index $i$, and let $\theta \sim \mathcal{N}(\mu, \sigma^2)$.

**Derivation for Criteria 3:** $\mathbb{E}_{q_\phi}|\theta|$    We have

$$\mathbb{E}_{q_\phi}|\theta| = \int_{-\infty}^{\infty} |x| \cdot \frac{1}{\sqrt{2\pi}\sigma} \exp\left(-\frac{(x-\mu)^2}{2\sigma^2}\right) dx.$$

Due to the symmetry of $\theta$, we can assume $\mu \geq 0$ without loss of generality. Letting $t = \frac{x-\mu}{\sigma}$, we have

$$\mathbb{E}_{q_\phi}|\theta| = \int_{-\infty}^{\infty} |\sigma t + \mu| \cdot \frac{1}{\sqrt{2\pi}} \exp\left(-\frac{t^2}{2}\right) dt$$

$$= -\int_{-\infty}^{-\frac{\mu}{\sigma}} (\sigma t + \mu) \cdot \frac{1}{\sqrt{2\pi}} \exp\left(-\frac{t^2}{2}\right) dt + \int_{-\frac{\mu}{\sigma}}^{\infty} (\sigma t + \mu) \cdot \frac{1}{\sqrt{2\pi}} \exp\left(-\frac{t^2}{2}\right) dt$$

$$= \sigma \left(-\int_{-\infty}^{-\frac{\mu}{\sigma}} t \cdot \frac{1}{\sqrt{2\pi}} \exp\left(-\frac{t^2}{2}\right) dt + \int_{-\frac{\mu}{\sigma}}^{\infty} t \cdot \frac{1}{\sqrt{2\pi}} \exp\left(-\frac{t^2}{2}\right) dt\right)$$

$$+ \mu \left(-\int_{-\infty}^{-\frac{\mu}{\sigma}} \frac{1}{\sqrt{2\pi}} \exp\left(-\frac{t^2}{2}\right) dt + \int_{-\frac{\mu}{\sigma}}^{\infty} \frac{1}{\sqrt{2\pi}} \exp\left(-\frac{t^2}{2}\right) dt\right).$$

Denote

$$\Phi(x) := \int_{-\infty}^{x} \frac{1}{\sqrt{2\pi}} \exp\left(-\frac{y^2}{2}\right) dy,$$

to be the Cumulative Distribution Function (CDF) of standard Gaussian distribution. We can easily verify

$$\Phi(x) + \Phi(-x) = 1.$$

Hence, we have

$$\mathbb{E}_{q_\phi}|\theta| = \sigma \left(\frac{1}{\sqrt{2\pi}} \exp\left(-\frac{t^2}{2}\right)\Big|_{-\infty}^{-\frac{\mu}{\sigma}} - \frac{1}{\sqrt{2\pi}} \exp\left(-\frac{t^2}{2}\right)\Big|_{-\frac{\mu}{\sigma}}^{\infty}\right) + \mu \left(-\Phi\left(-\frac{\mu}{\sigma}\right) + \Phi\left(\frac{\mu}{\sigma}\right)\right)$$

$$= \sigma \sqrt{\frac{2}{\pi}} \exp\left(-\frac{\mu^2}{\sigma^2}\right) + \mu \left(2\phi\left(\frac{\mu}{\sigma}\right) - 1\right). \tag{15}$$

Hence, we get the explicit form of criteria 3.

**Derivation for Criteria 5 and 6:** $\mathbb{E}_{q_\phi} e^{\lambda|\theta|}$ **and** $\mathrm{SNR}_{q_\phi}(e^{\lambda|\theta|})$    For $\lambda > 0$, setting $t = \frac{x-\mu}{\sigma}$, we have

$$\mathbb{E}_{q_\phi} e^{\lambda|\theta|} = \int_{-\infty}^{\infty} \exp(\lambda|x|) \frac{1}{\sqrt{2\pi}\sigma} \exp\left(-\frac{(x-\mu)^2}{2\sigma^2}\right) dx$$

$$= \int_{-\infty}^{\infty} \exp(\lambda|\mu + \sigma t|) \frac{1}{\sqrt{2\pi}} \exp\left(-\frac{t^2}{2}\right) dt$$

$$= \int_{-\infty}^{-\frac{\mu}{\sigma}} \exp(-\lambda(\mu + \sigma t)) \frac{1}{\sqrt{2\pi}} \exp\left(-\frac{t^2}{2}\right) dt + \int_{-\frac{\mu}{\sigma}}^{\infty} \exp(\lambda(\mu + \sigma t)) \frac{1}{\sqrt{2\pi}} \exp\left(-\frac{t^2}{2}\right) dt$$

$$= \int_{-\infty}^{-\frac{\mu}{\sigma}} \frac{1}{\sqrt{2\pi}} \exp\left[-\left(\frac{t^2}{2} + \lambda\sigma t + \lambda\mu\right)\right] dt + \int_{-\frac{\mu}{\sigma}}^{\infty} \frac{1}{\sqrt{2\pi}} \exp\left[-\left(\frac{t^2}{2} - \lambda\sigma t - \lambda\mu\right)\right] dt$$

$$= \int_{-\infty}^{-\frac{\mu}{\sigma}} \frac{1}{\sqrt{2\pi}} \exp\left[-\frac{(t + \lambda\sigma)^2}{2}\right] dt \cdot \exp\left(\frac{\lambda^2\sigma^2}{2} - \lambda\mu\right)$$

$$+ \int_{-\frac{\mu}{\sigma}}^{\infty} \frac{1}{\sqrt{2\pi}} \exp\left[-\frac{(t - \lambda\sigma)^2}{2}\right] dt \cdot \exp\left(\frac{\lambda^2\sigma^2}{2} + \lambda\mu\right)$$

$$= \Phi\left(-\frac{\mu}{\sigma} + \lambda\sigma\right) e^{\frac{\lambda^2\sigma^2}{2} - \lambda\mu} + \Phi\left(\frac{\mu}{\sigma} + \lambda\sigma\right) e^{\frac{\lambda^2\sigma^2}{2} + \lambda\mu}. \tag{16}$$

For $\mathrm{SNR}_{q_\phi}(e^{\lambda|\theta|})$, notice that

$$\mathrm{Var}_{q_\phi} e^{\lambda|\theta|} = \mathbb{E}_{q_\phi}(e^{2\lambda|\theta|}) - \left(\mathbb{E}_{q_\phi} e^{\lambda|\theta|}\right)^2.$$

Given that we already have the expectation $\mathbb{E}_{q_\phi} e^{\lambda|\theta|}$, we can readily compute the first term of $\mathrm{Var}_{q_\phi}\left(e^{\lambda|\theta|}\right)$, by substituting $\lambda$ with $2\lambda$. This allows us to determine $\mathrm{SNR}_{q_\phi}(e^{\lambda|\theta|})$, completing the proof.

## C TRAINING DETAILS

For training, we solve the optimization problem (3) following the procedure in Algorithm 1. For a fair comparison, we train our algorithms for 200 epochs, with a batch size 128, SGD optimizer with an initial learning rate of 0.01, and a Cosine annealing decay for learning rate following Kong et al. (2023). Since we cannot determine the final sparsity of previous methods before training, we conduct the training process with previous methods first, record the final sparsity achieved, and then set a similar value for SSVI to ensure an equitable comparison.

## D ADDITIONAL EXPERIMENTS

In this section, we undertake further experiments addressing two distinct types of distribution shift challenges to show the ability of uncertainty. This encompasses standard assessments with Out-of-Distribution (OOD) datasets and evaluations on datasets subjected to various corruptions. Additionally, we carry out additional ablation studies to examine the disparities among the five different removal criteria proposed in Section 3.3.1.

### D.1 MORE DISTRIBUTION SHIFT TASKS

**OOD detection** We train utilizing SSVI on CIFAR-10/100 and subsequently test on CIFAR-100/10 and SVHN, in line with the methodologies established in prior research (Ritter et al., 2021). We calculated both the area under the receiver operating characteristic (AUROC) and the area under the precision-recall curve (AUPR) using our models. In this context, we employed the mBNN algorithm (Kong et al., 2023), the strongest previous work, as our baseline, ensuring equal sparsity for a fair and meaningful comparison.

| Train set | CIFAR-10 | | | | CIFAR-100 | | | |
|---|---|---|---|---|---|---|---|---|
| Test set | CIFAR-100 | | SVHN | | CIFAR-10 | | SVHN | |
| Metric | AUROC | AUPR | AUROC | AUPR | AUROC | AUPR | AUROC | AUPR |
| mBNN | 0.84 | 0.81 | 0.89 | 0.94 | 0.76 | 0.72 | 0.78 | 0.88 |
| SSVI (ours) | **0.86** | **0.84** | **0.93** | **0.96** | **0.80** | **0.77** | **0.85** | **0.91** |

Table 3: Results for OOD detection. SSVI demonstrates superior performance across all metrics.

**Corrupted datasets** We evaluate our model, which has been trained using SSVI on CIFAR-10/100, against their corrupted counterparts, CIFAR-10-C and CIFAR-100-C, as introduced by Hendrycks & Dietterich (2019). These datasets feature 15 unique types of corruptions, each with 5 levels of severity. For each level of corruption, we calculate the average results across all 15 corruption types. The outcomes of this evaluation are presented in Figure 4. As a comparative standard, we continue to use mBNN (Kong et al., 2023) as the baseline. The findings indicate that SSVI consistently surpasses mBNN across all evaluated metrics.

### D.2 ADDITIONAL ABLATION STUDIES

We introduce various criteria for weight removal in Section 3.3.1, which are utilized to identify the top-$K_t$ weights at each removal step $t$. To assess the differences among the first 5 criteria outlined in Section 3.3.1, we calculate the ratio of intersection over union (IoU) for all 10 possible pairings between these criteria. In Figure 5, we display the average IoU scores across these pairs over the course of the removal steps. Our results show that although the criteria initially exhibit a high degree of similarity, indicated by high IoU scores, they begin to diverge as the training advances. This divergence is likely attributed to the increasing precision of uncertainty information, leading to more refined outcomes.

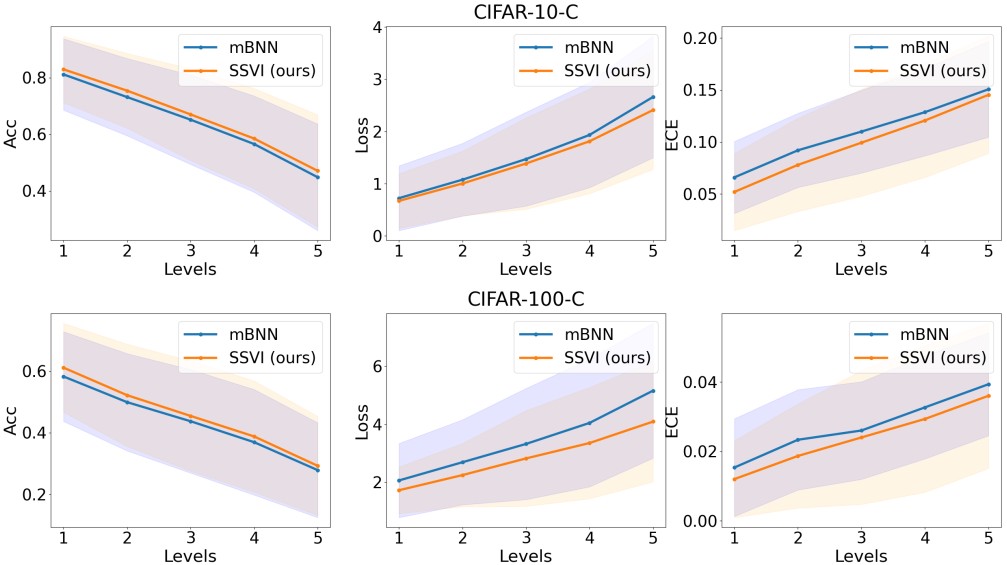

Figure 4: Performance analysis on CIFAR-10-C and CIFAR-100-C. SSVI shows consistently better results across all metrics under corruption.

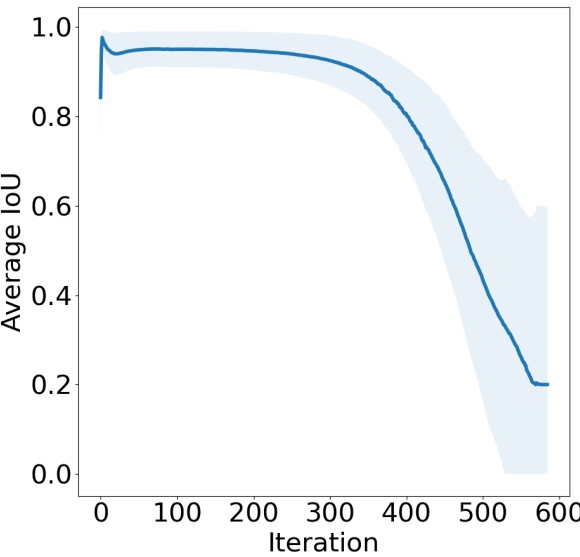

Figure 5: Average IoU of all 10 pairs between 5 different criteria.

For a detailed examination of the IoU scores among various pairs, we present a pair-wise matrix of the five scores at the start, midpoint, and conclusion of the training in Table 4. This analysis not only reinforces our earlier observations from Figure 5 but also highlights that traditional criteria, specifically $|\mu|$ and $\mathrm{SNR}_{q_\phi}(\theta)$, exhibit a persistent similarity during the entire training period. On the other hand, the three novel criteria we introduced: $\mathbb{E}_{q_\phi}|\theta|$, $\mathrm{SNR}_{q_\phi}(|\theta|)$, and $\mathbb{E}_{q_\phi}e^{\lambda|\theta|}$, show a marked deviation from these conventional metrics, emphasizing the innovative aspect of our methodology.

Table 4: IoU matrices in different training stages.

|  | $\|\boldsymbol{\mu}\|$ | $\mathbf{SNR}_{q_\phi}(\boldsymbol{\theta})$ | $\mathbb{E}_{q_\phi}\|\boldsymbol{\theta}\|$ | $\mathbf{SNR}_{q_\phi}(\|\boldsymbol{\theta}\|)$ | $\mathbb{E}_{q_\phi}e^{\lambda\|\boldsymbol{\theta}\|}$ |
|---|---|---|---|---|---|
| $\|\boldsymbol{\mu}\|$ | 1.0000 | 0.9995 | 0.9122 | 0.9981 | 0.9122 |
| $\mathbf{SNR}_{q_\phi}(\boldsymbol{\theta})$ | 0.9995 | 1.0000 | 0.9118 | 0.9985 | 0.9118 |
| $\mathbb{E}_{q_\phi}\|\boldsymbol{\theta}\|$ | 0.9122 | 0.9118 | 1.0000 | 0.9118 | 0.9996 |
| $\mathbf{SNR}_{q_\phi}(\|\boldsymbol{\theta}\|)$ | 0.9981 | 0.9985 | 0.9118 | 1.0000 | 0.9118 |
| $\mathbb{E}_{q_\phi}e^{\lambda\|\boldsymbol{\theta}\|}$ | 0.9122 | 0.9118 | 0.9996 | 0.9118 | 1.0000 |

Table 5: IoU matrix at the beginning.

|  | $\|\boldsymbol{\mu}\|$ | $\mathbf{SNR}_{q_\phi}(\boldsymbol{\theta})$ | $\mathbb{E}_{q_\phi}\|\boldsymbol{\theta}\|$ | $\mathbf{SNR}_{q_\phi}(\|\boldsymbol{\theta}\|)$ | $\mathbb{E}_{q_\phi}e^{\lambda\|\boldsymbol{\theta}\|}$ |
|---|---|---|---|---|---|
| $\|\boldsymbol{\mu}\|$ | 1.0000 | 0.9999 | 0.7723 | 0.8177 | 0.7722 |
| $\mathbf{SNR}_{q_\phi}(\boldsymbol{\theta})$ | 0.9999 | 1.0000 | 0.7722 | 0.8177 | 0.7722 |
| $\mathbb{E}_{q_\phi}\|\boldsymbol{\theta}\|$ | 0.7723 | 0.7722 | 1.0000 | 0.6786 | 0.9895 |
| $\mathbf{SNR}_{q_\phi}(\|\boldsymbol{\theta}\|)$ | 0.8177 | 0.8177 | 0.6786 | 1.0000 | 0.6790 |
| $\mathbb{E}_{q_\phi}e^{\lambda\|\boldsymbol{\theta}\|}$ | 0.7722 | 0.7722 | 0.9895 | 0.6790 | 1.0000 |

Table 6: IoU matrix in the middle.

|  | $\|\boldsymbol{\mu}\|$ | $\mathbf{SNR}_{q_\phi}(\boldsymbol{\theta})$ | $\mathbb{E}_{q_\phi}\|\boldsymbol{\theta}\|$ | $\mathbf{SNR}_{q_\phi}(\|\boldsymbol{\theta}\|)$ | $\mathbb{E}_{q_\phi}e^{\lambda\|\boldsymbol{\theta}\|}$ |
|---|---|---|---|---|---|
| $\|\boldsymbol{\mu}\|$ | 1.0000 | 0.9998 | 0.1528 | 0.0829 | 0.1190 |
| $\mathbf{SNR}_{q_\phi}(\boldsymbol{\theta})$ | 0.9998 | 1.0000 | 0.1528 | 0.0829 | 0.1190 |
| $\mathbb{E}_{q_\phi}\|\boldsymbol{\theta}\|$ | 0.1528 | 0.1528 | 1.0000 | 0.0389 | 0.8501 |
| $\mathbf{SNR}_{q_\phi}(\|\boldsymbol{\theta}\|)$ | 0.0829 | 0.0829 | 0.0389 | 1.0000 | 0.0374 |
| $\mathbb{E}_{q_\phi}e^{\lambda\|\boldsymbol{\theta}\|}$ | 0.1190 | 0.1190 | 0.8501 | 0.0374 | 1.0000 |

Table 7: IoU matrix at the end.