# OpenReview forum: "Training Bayesian Neural Networks with Sparse Subspace Variational Inference"
_ICLR.cc/2024/Conference — ICLR 2024 poster_

### Official Review · Reviewer_SCNX · 2023-10-23

**Soundness:** 3 good
**Presentation:** 2 fair
**Contribution:** 3 good
**Rating:** 6
**Confidence:** 4

**Summary:**

The paper proposes a procedure for training sparse Bayesian neural networks with a factorized variational distribution. It does so by alternating between optimizing the parameters of a subset of the weight distributions, with the remaining ones zeroed out, and updating the subset of non-zero weights by adding to and removing from that subset based on a range of criteria. The experiments show improved accuracy and calibration on CIFAR10/100 over methods from the literature at higher sparsity.

Overall, I would say that the paper introduces some reasonable but not ground-breaking technical material, however it is let down by a severely limited evaluation, omitting most standard uncertainty estimation benchmarks. With some further issues around overclaiming novelty, I would argue for rejecting the paper in its current form.

POST REBUTTAL UPDATE:
Thank you for the extensive updates. These largely address my core concerns (limited evaluation, lack of demonstration of difference between pruning scores), so I will raise my score. I would suggest at least referencing Appendix D from the main text and if possible moving some figures to the experiment section.

I did not see an update to the paper reflecting W4. I completely agree that the method has limitations, my point is only that it is not a post-hoc pruning approach and the paper shouldn't misrepresent this.

**Strengths:**

- the paper is overall clear in what it does and why. It shouldn't be too difficult to implement the method based on the description.
- selection criteria for weight pruning may be of independent interest.
- there is a decent range of ablation studies on various aspects of the proposed method.
- performance on the datasets and metrics under consideration is better than that of the baselines.

**Weaknesses:**

1. the experiments are quite bare-bones for a BNN paper, there is no evaluation of predictive uncertainty besides calibration -- we don't need a Bayesian approach do well on this metric. I would either suggest adding e.g. a temperature scaling baseline applied to a sparse deterministic net or (preferably) the usual out-of-distribution and distribution shift benchmarks.
2. primarily testing at a single sparsity level as in Table 2 also seems a bit limited to me. In my view, there are broadly two possible goals when using sparsity: opitimizing sparsity at a given performance level, e.g. close to optimal, or optimizing performance at a given sparsity level. I would have liked to see more figures in the style of Figure 2 left and Figure 3 to cover both of these settings also for the baselines.
3. I would have liked to see a bit more in-depth investigation of the pruning criteria, e.g. a plot of Spearman correlations between the preferred score and the others throughout training or a correlation matrix at various stages (say beginning, halfway through and end of training). I must say that I am not overly convinced that they matter too much, the variation of accuracy in Fig 2 seems to be only about 0.5% (although see questions). So I think it might be worth saving the page discussing the criteria in fewer of more thorough experiments.
4. the paper makes some rather inaccurate claims vs the existing literature. In particular, it is not the first paper introducing a "fully sparse BNN framework that maintains a consistently sparse Bayesian model through- out the training and inference", this statement also applies to the (Ritter et al., 2021) paper, which is incorrectly cited as a post-hoc pruning paper (the paper does use post-hoc pruning as an optional step to further increase sparsity, but the core low-rank parameterization is maintained throughout training). This doesn't affect the contribution of course, but prior work needs to be contextualized correctly.
5. I don't really see the need to make such claims in the first place, it is not obvious that sparsity in training is desirable. Of course it may be the case that a larger network that would not fit into memory without sparsity performs better, but then this needs to be demonstrated (or like-wise any hypothetical training speed increases resulting from a reduced number of FLOPs - in the age of parallelized computation, that is a mostly meaningless metric if it cannot be shown that a practical implementation can lead to actual cost savings).
6. the abstract is simultaneously wordy and vague. I did not know what the paper was doing specifically after reading it, even though it really isn't hard to describe the method in 1 or 2 sentences. I would say that the low-rank/basis terminology led me in the wrong direction of thinking and a pruning-based description would have been clearer, but this may of course differ for readers with a different background.

**Questions:**

- how are the mean values initialized for weights that are added to the active set? I assume 0? Do you reuse the old value if a weight had been temporarily removed?
- I'm not sure I follow the discussion of Figure 2 right. For the initialization of the variance of a mean-field Gaussian, identical considerations apply as for the initialization of weights in deterministic networks, so loosely speaking we want to scale the sum of variance of the means and the initial value of the variance parameter inversely with the number of units to avoid the activations diverging with increasing depth. So to me it seems natural, that dense VI would reach this critical threshold before a pruned variance, as the latter is simply removing terms from a positive sum and thus decreasing variance. Am I missing something/misinterpreting the figure?
- Can you demonstrate any practical benefits from sparsity during training/the FLOP reduction translating to real time speedups?

For suggestions see weaknesses.

---

> ### Author Response · Authors · 2023-11-21
> **Response Part 1**
>
> We greatly appreciate the extensive comments. We have carefully addressed each of the weaknesses and questions raised in your feedback. In light of these comprehensive revisions, we kindly hope that you will consider raising your score.
>
> ## Experiments about OOD detection and distribution shift benchmark
>
> We extended our experiments to two distribution shift benchmarks including both Out-of-Distribution (OOD) detection tasks and robustness benchmarks. The results demonstrate that our SSVI consistently improves over the strongest baseline, mBNN, significantly across all sparsity levels, metrics, and tasks. The results are presented both in the subsequent sections and in the updated Appendix D.
>
> For OOD detection tasks, we computed the area under the receiver operating characteristic (AUROC) and the area under the precision-recall curve (AUPR) using our models, which were trained using SSVI on CIFAR-10/100 and tested on CIFAR-100/10 and SVHN, following the methodology of previous studies [1]. In this context, we employed the mBNN algorithm, the strongest previous work, as our baseline, ensuring equal sparsity for a fair and meaningful comparison.
>
> |  **Train set**  |  |           |   CIFAR-10   |       |      | |          |   CIFAR-100   |       |      |
> |:-----------:|:--------:|:---------:|:----:|:-----:|:----:|:---------:|:--------:|:----:|:-----:|:----:|
> |   Test set  | | CIFAR-100 |      |  SVHN |      |  | CIFAR-10 |      |  SVHN |      |
> |       Metric      |    Sparsity      |   AUROC   | AUPR | AUROC | AUPR |    Sparsity       |   AUROC  | AUPR | AUROC | AUPR |
> |     mBNN    |   96.4%  |    0.84   | 0.81 |  0.89 | 0.94 |   87.8%   |   0.76   | 0.72 |  0.78 | 0.88 |
> | SSVI (ours) |    95%   |    **0.86**   | **0.84** |  **0.93** | **0.96** |    **90%**    |   **0.80**   | **0.77** |  **0.85** | **0.91** |
>
>
> We proceed to assess our model, trained using SSVI on CIFAR-10/100, on the respective corrupted versions, CIFAR-10/100-C. Both datasets encompass 15 distinct corruptions, each classified into 5 corruption levels. The outcomes are presented in the following table, illustrating curves for the average accuracy, loss, and Expected Calibration Error (ECE) across all 15 different corruptions for each level.
>
> | Dataset       | Model              | Level 1 | Level 2 | Level 3 | Level 4 | Level 5 |
> |---------------|--------------------|---------|---------|---------|---------|---------|
> | CIFAR-10-C    | mBNN (ACC)         | 0.811   | 0.731   | 0.651   | 0.565   | 0.448   |
> |               | **SSVI (ACC)**     | **0.829**   | **0.754**   | **0.670**    | **0.585**   | **0.471**   |
> |               | mBNN (Loss)        | 0.721   | 1.075   | 1.469   | 1.931   | 2.659   |
> |               | **SSVI (Loss)**    | **0.668**   | **1.001**   | **1.384**   | **1.810**    | **2.408**   |
> |               | mBNN (ECE Loss)    | 0.066   | 0.092   | 0.110    | 0.129   | 0.151   |
> |               | **SSVI (ECE Loss)**| **0.052**   | **0.078**   | **0.099**   | **0.121**   | **0.145**   |
> |---------------|--------------------|---------|---------|---------|---------|---------|
> | CIFAR-100-C   | mBNN (ACC)         | 0.583   | 0.499   | 0.437   | 0.369   | 0.279   |
> |               | **SSVI (ACC)**     | **0.611**   | **0.522**   | **0.455**   | **0.388**   | **0.293**   |
> |               | mBNN (Loss)        | 2.062   | 2.69    | 3.320    | 4.045   | 5.156   |
> |               | **SSVI (Loss)**    | **1.725**   | **2.247**   | **2.821**   | **3.351**   | **4.090**    |
> |               | mBNN (ECE Loss)    | 0.015   | 0.023   | 0.026   | 0.033   | 0.039   |
> |               | **SSVI (ECE Loss)**| **0.012**   | **0.019**   | **0.024**   | **0.029**   | **0.036**   |
> |---------------|--------------------|---------|---------|---------|---------|---------|
>
>
> ## Calibration metric
>
> While we understand your perspective, we respectfully disagree with the notion that calibration is not valuable. Calibration is widely recognized as a crucial metric in Bayesian deep learning literature, particularly as an indicator of a model's uncertainty capabilities. This perspective is not only supported in the paper you referred to [1] but also in our most closely related baseline [8]. Additionally, calibration is considered a significant metric in various studies investigating the posterior of Bayesian models, as seen in references [9, 10].

---

> ### Author Response · Authors · 2023-11-21
> **Response Part 2**
>
> ## Results of more sparsity levels for baseline algorithms
>
> For our SSVI, we present outcomes at various sparsity levels in Figure 2, as you noticed. However, this does not extend to the baseline algorithms. Actually, all the baseline algorithms utilize implicit regularization to steer the model towards a level of sparsity that is not predefined, leading to a process that is frequently unstable and heavily dependent on precise hyperparameter tuning to achieve a satisfactory level of sparsity. Therefore, we first train the baselines following the settings in their original papers, then adopt the obtained sparsity level in our SSVI training to ensure a fair comparison.
>
> We want to highlight that this is one of the advantages of our SSVI —- the sparsity can be flexibly assigned before training. We demonstrate the efficacy of SSVI across a range of sparsity levels and also explore its performance with various optimization schedules within the same sparsity level, as shown in Figures 2 and 3.
>
> ## Influence of pruning criteria
>
> Indeed, the choice of pruning criteria is crucial. It is important to note that global metrics like Spearman correlations may not be the most suitable metric for assessing the differences between various scores in this context. Since we utilize the score solely for selecting the top-K weights without incorporating it into the training process, and considering that K is significantly smaller than the total number of weights, a high Spearman score may not accurately reflect the divergence in the chosen weights.
>
> Therefore, we have opted for the ratio of intersection over union (IoU). Each score selects the top-K weights for a fixed number K, and we calculate the intersection over union for all pairs among our five different scores.
>
> In the following, we present the Intersection over Union (IoU) matrices for the five pruning criteria at the initial, middle, and final stages of training, respectively. Additionally, we include in the supplementary material the average of all 10 IoU values for different pairs across the training steps. The findings reveal that while the criteria show high similarity (reflected in high IoU scores) in the initial phase, they diverge as training progresses, potentially due to more precise uncertainty information and consequently more appropriate criteria. It is also notable that the traditional criteria, specifically $|\mu|$ and $\mathrm{SNR}{q_{\phi}}(\theta)$, exhibit a persistent similarity during the entire training period. On the other hand, the three novel criteria we introduced: $\mathbb{E}{q_\phi}|\theta|$, $\mathrm{SNR}{q_{\phi}}(|\theta|)$, and $\mathbb{E}{q_\phi}e^{\lambda|\theta|}$, show a marked deviation from these conventional metrics, emphasizing the innovative aspect of our methodology.
>
>
> —-----------------IoU matrix at the beginning of training—-----------------
>
> |                                          | $\|\mu\|$ | $SNR_{q_{\phi}}(\theta)$ | $\mathbb{E}_{q\phi}\|\theta\|$ | $SNR_{q_{\phi}}(\|\theta\|)$ | $\mathbb{E}_{q\phi}e^{\lambda\|\theta\|}$ |
> |------------------------------------------|-----------|--------------------------|--------------------------------|------------------------------|------------------------------------------|
> | $\|\mu\|$                                | 1.0       | 0.9995                   | 0.9122                         | 0.9981                       | 0.9122                                   |
> | $SNR_{q_{\phi}}(\theta)$                 | 0.9995    | 1.0                      | 0.9118                         | 0.9985                       | 0.9118                                   |
> | $\mathbb{E}_{q\phi}\|\theta\|$            | 0.9122    | 0.9118                   | 1.0                            | 0.9118                       | 0.9996                                   |
> | $SNR_{q_{\phi}}(\|\theta\|)$             | 0.9981    | 0.9985                   | 0.9118                         | 1.0                          | 0.9118                                   |
> | $\mathbb{E}_{q\phi}e^{\lambda\|\theta\|}$ | 0.9122    | 0.9118                   | 0.9996                         | 0.9118                       | 1.0                                      |

---

> ### Author Response · Authors · 2023-11-21
> **Response Part 3**
>
> —----------------IoU matrix at the middle of training—----------------
>
> |                                          | $\|\mu\|$ | $SNR_{q_{\phi}}(\theta)$ | $\mathbb{E}_{q\phi}\|\theta\|$ | $SNR_{q_{\phi}}(\|\theta\|)$ | $\mathbb{E}_{q\phi}e^{\lambda\|\theta\|}$ |
> |------------------------------------------|-----------|--------------------------|--------------------------------|------------------------------|------------------------------------------|
> | $\|\mu\|$                                | 1.0       | 0.9999                   | 0.7723                         | 0.8177                       | 0.7722                                   |
> | $SNR_{q_{\phi}}(\theta)$                 | 0.9999    | 1.0                      | 0.7722                         | 0.8177                       | 0.7722                                   |
> | $\mathbb{E}_{q\phi}\|\theta\|$            | 0.7723    | 0.7722                   | 1.0                            | 0.6786                       | 0.9895                                   |
> | $SNR_{q_{\phi}}(\|\theta\|)$             | 0.8177    | 0.8177                   | 0.6786                         | 1.0                          | 0.6790                                   |
> | $\mathbb{E}_{q\phi}e^{\lambda\|\theta\|}$ | 0.7722    | 0.7722                   | 0.9895                         | 0.6790                       | 1.0                                      |
>
> —-----------------IoU matrix at the end of training—-----------------
>
> |                                          | $\|\mu\|$ | $SNR_{q_{\phi}}(\theta)$ | $\mathbb{E}_{q\phi}\|\theta\|$ | $SNR_{q_{\phi}}(\|\theta\|)$ | $\mathbb{E}_{q\phi}e^{\lambda\|\theta\|}$ |
> |------------------------------------------|-----------|--------------------------|--------------------------------|------------------------------|------------------------------------------|
> | $\|\mu\|$                                | 1.0       | 0.9998                   | 0.1528                         | 0.0829                       | 0.1190                                   |
> | $SNR_{q_{\phi}}(\theta)$                 | 0.9998    | 1.0                      | 0.1528                         | 0.0829                       | 0.1190                                   |
> | $\mathbb{E}_{q\phi}\|\theta\|$            | 0.1528    | 0.1528                   | 1.0                            | 0.0389                       | 0.8501                                   |
> | $SNR_{q_{\phi}}(\|\theta\|)$             | 0.0829    | 0.0829                   | 0.0389                         | 1.0                          | 0.0374                                   |
> | $\mathbb{E}_{q\phi}e^{\lambda\|\theta\|}$ | 0.1190    | 0.1190                   | 0.8501                         | 0.0374                       | 1.0                                      |
>
> ## Comparison with Ritter et al. 2021
>
> Although (Ritter et al., 2021) achieve parameter reduction by storing the inducing weights $U$ that have much lower dimensions, they still require sampling full-dimension original weights $W$ in each forward pass, as stated in Section 3.2 of their paper. As a result, their computation costs in both training and inference remain similar or even larger than the dense BNNs. In contrast, our method directly enforces sparsity in $W$, which has a substantial reduction of computation costs theoretically. Besides, the sampling rule in (Ritter et al., 2021) requires a Gaussian parameterization, whereas our method is more flexible and can generalize beyond Gaussian parametrization. Yet, we still thank you for the comment and will change the description of (Ritter et al., 2021) in the revision.
>
> ## Why do we need sparsity in training
>
> We note that the concern raised is not unique to our work on sparse Bayesian Neural Networks (BNNs) but extends to the broader field of sparse neural networks. The advantages of having sparsity in training have been addressed in previous discussions, such as [5]. Here we formulate the answer about why we need sparsity in training from two directions: speedup and performance.
>
> **Real-time speedup:**
>
> First, we want to clarify that sparsity in training can lead to actual, real-time performance improvements on modern hardware, beyond just theoretical advantages suggested by FLOPs reduction. As shown by [5], *“Another notable success was recently demonstrated by DeepSparse [11, 12], which successfully deploys large-scale BERT-level sparse models on modern Intel CPUs, obtaining 10× model size compression with < 1% accuracy drop, 10× CPU-inference speedup with < 2% drop, and 29× CPU-inference speedup with < 7.5% drop.”* Additionally, there are considerable efforts, such as those by NVIDIA, to develop advanced GPU kernels like Sputnik [13] and the NVIDIA Ampere Architecture [14], which *“built the momentum to better support finer-grained sparsity”* and thereby increase real-time speedup. We will explore how to achieve real wall-time speedup with the specific hardware in our future work.

---

> ### Author Response · Authors · 2023-11-21
> **Response Part 4**
>
> **Performance improvement:**
>
> As highlighted in your review, sparsity is key to training larger models within memory constraints. This leads to the question: Why not simply opt for a small dense model instead? The answer lies in the distinct advantages of large sparse models. Also shown by [5], *“Sparse NNs keep the same model architectures with those dense large models, but only activate a small fraction of weights. Since state-of-the-art sparse NNs can match the utility performance of their dense counterparts (through either pruning or DST/PaI), sparsity can potentially be a free lunch to make training/inference cheaper while performing the same well.”* This is also illustrated in our research, particularly evident in Figure 2 where we see a largely stable performance curve against sparsity level and hyperparameter selection. In contrast, a small dense model with a similar parameter count typically exhibits a noticeable drop in performance when compared to its larger, dense equivalent, as demonstrated in references [2,3,4]. Consequently, when considering models with comparable parameters, the superior performance of large sparse models makes them a preferable choice over small dense models in training scenarios.
>
>
> A more contemporary and well-known instance highlighting the effectiveness of large sparse models is the adoption of Mixture-of-Expert (MoE) techniques. These techniques employ large sparse models not just during training but also throughout the inference process. They have been recognized as optimal solutions at the industrial level, as evidenced in high-profile projects such as Switch transformers [6] and Glam [7]. This trend underscores the practicality and efficiency of large sparse models in advanced applications, demonstrating their utility beyond the theoretical advantages of reduced FLOPs.
>
>
>
> ## Abstract presentation
>
> We would like to clarify that our paper is not only about pruning. Our paper is about how to find an optimal sparse subspace to do variational inference. To solve this problem, we optimize the sparse subspace and the corresponding variational model together. In this context, a removal(pruning)-based algorithm emerges as a natural solution within an alternative optimization schedule. We appreciate your input and will refine the abstract to provide a clearer description in the revised draft.
>
> ## Initialization of weight mean
>
> For initializing the mean values of weights added to the active set, we use a zero initialization.
>
>
> ## About Figure 2 right
>
> Your interpretation is essentially accurate. The phenomenon you described underscores a key advantage of our SSVI over conventional VI in Bayesian Neural Networks (BNNs).
> In Figure 2, we empirically verified this interpretation, as you correctly noted.
> It is important to recognize that training BNNs is more sensitive to initialization than training conventional neural networks. This makes our comparison between SSVI and VI particularly revealing. Consequently, the findings presented in Figure 2 right are neither trivial nor obvious and should be regarded as substantial.

---

> ### Author Response · Authors · 2023-11-21
> **Response Part 5**
>
> ## References
>
> [1] Hippolyt Ritter, Martin Kukla, Cheng Zhang, and Yingzhen Li. Sparse uncertainty representation in deep learning with inducing weights. Advances in Neural Information Processing Systems, 34: 6515–6528, 2021.
>
> [2] Li, Zhuohan, et al. "Train big, then compress: Rethinking model size for efficient training and inference of transformers." International Conference on machine learning. PMLR, 2020.
>
> [3] ​​Evci, Utku, et al. "Rigging the lottery: Making all tickets winners." International Conference on Machine Learning. PMLR, 2020.
>
> [4] Liu, Shiwei, et al. "The unreasonable effectiveness of random pruning: Return of the most naive baseline for sparse training." arXiv preprint arXiv:2202.02643 (2022).
>
> [5] Liu, Shiwei, and Zhangyang Wang. "Ten lessons we have learned in the new" sparseland": A short handbook for sparse neural network researchers." arXiv preprint arXiv:2302.02596 (2023).
>
> [6] Fedus, William, Barret Zoph, and Noam Shazeer. "Switch transformers: Scaling to trillion parameter models with simple and efficient sparsity." The Journal of Machine Learning Research 23.1 (2022): 5232-5270.
>
> [7] Du, Nan, et al. "Glam: Efficient scaling of language models with mixture-of-experts." International Conference on Machine Learning. PMLR, 2022.
>
> [8] Kong, Insung, et al. "Masked Bayesian Neural Networks: Theoretical Guarantee and its Posterior Inference." arXiv preprint arXiv:2305.14765 (2023).
>
> [9] Wenzel, Florian, et al. "How good is the bayes posterior in deep neural networks really?." ICML 2020.
>
> [10] Izmailov, Pavel, et al. "What are Bayesian neural network posteriors really like?." International conference on machine learning. PMLR, 2021.
>
> [11] Kurtz, Mark, et al. "Inducing and exploiting activation sparsity for fast inference on deep neural networks." International Conference on Machine Learning. PMLR, 2020.
>
> [12] Kurtic, Eldar, et al. "The optimal bert surgeon: Scalable and accurate second-order pruning for large language models." arXiv preprint arXiv:2203.07259 (2022).
>
> [13] Gale, Trevor, et al. "Sparse gpu kernels for deep learning." SC20: International
> Conference for High Performance Computing, Networking, Storage and Analysis. IEEE, 2020.
>
> [14] 2020 Nvidia a100 tensor core gpu architecture. https://www.nvidia.com/content/dam/en- zz/Solutions/Data-Center/nvidia-ampere-architecture-whitepaper.pdf .

---

> ### Author Response · Authors · 2023-11-22
> **Thank you once again for your comprehensive review of our paper!**
>
> Thank you once again for your comprehensive review of our paper. We have addressed each of your comments in the five parts above and are confident that our responses have resolved all your concerns. As the close of the discussion period nears, we welcome any further questions you may have or would appreciate if you consider increasing the score for our submission. We are grateful for the time and effort you have invested in this review process!

---

### Official Review · Reviewer_KZuX · 2023-10-31

**Soundness:** 3 good
**Presentation:** 3 good
**Contribution:** 4 excellent
**Rating:** 8
**Confidence:** 3

**Summary:**

The paper proposes a novel inference method for sparse Bayesian neural networks based on variational inference, namely Sparse Subspace Variational Inference (SSVI). The idea, that is different to the previous sparse BNNs, is to explicitly keep the number of non-zero weights in the neural network fixed while updating which weights are included or excluded as non-significant. This is in contrast to the existing literature, where sparsity is induced by special priors and, firstly, still starts with a dense model that is sparsified during training, which still make inference during training computationally expensive. Secondly, sparsity-inducing priors do not allow to set a specific level of sparsity, which may require a user to play with hyperparameters of the prior to get the desired level of sparsity. The proposed method has a level of sparsity as a hyperparameter and it is fixed from the start of the training, saving computational efforts from the start. Experiments in CIFAR-10/100 show the superiority of the proposed method in comparison to other sparse NNs in accuracy, loss, ECE, and training FLOPs.

**Strengths:**

* Interesting new idea for sparse BNNs, that:
* ... is shown to be working empirically
* ... has a solid theoretical model that is intuitive
* ... brings significant benefits in terms of computational cost

Originality: The idea presented in the paper appears to be novel.

Quality: The proposed method is well defined and all the inference steps seem to be sound. The experiments are well designed and executed. Ablation study is presented to different aspects of the method.

Clarity: The paper is very well written and easy to follow.

Significance: I believe the paper is of extreme interest for the Bayesian deep learning community as it addresses the problem of scaling inference of BNNs that is a known issue of the concept-appealing approach.

**Weaknesses:**

All of the below is not major weakness points.

Originality: A bit of context in terms of BNNs is missing. The review of the existing methods solely (though understandably) focusses on sparse models, leaving behind other approaches of making efficient Bayesian inference for NNs. E.g., Cobb, A.D. and Jalaian, B., 2021. Scaling Hamiltonian Monte Carlo inference for Bayesian neural networks with symmetric splitting. In Uncertainty in Artificial Intelligence (pp. 675-685). PMLR.

Quality: The empirical evaluation of the model though done rather thoroughly but only on 2 medium-size/complexity datasets from the same domain on 1 architecture. It would be interesting to see more experiments.

Clarity: Though the model is mostly well-written, there are some moments that are missing. See details below.

Specific comments:
1.	Abstract. 20x compression, 20x FLOPs reduction – in comparison to what?
2.	Abstract. “surpassing VI-trained dense BNNs” – in terms of what?
3.	Figure 1. Not defined names of baselines.
4.	Eq. (3) it is better to include explicitly what is the operator in the last 2 equations
5.	Around eq. (4). What are p, B, q, W?
6.	Eq. (5). What is l?
7.	Table 1. What is the difference between row 2 and 3 and then 5 and 6?
8.	It is unclear about eq. (11) and (12) (from the appendix), which one is used at the end?

Minor:
1.	Ablation study on updating \phi. ‘… in Figure 3. the optimal …’ -> The

**Questions:**

I like the paper, my minor weakness findings are listed above, but there is nothing the authors should or could address during the rebuttal.

---

> ### Author Response · Authors · 2023-11-21
> **Response Part 1**
>
> Thank you for your supportive and thoughtful comments! We are grateful for your acknowledgment of the significance, novelty, and potential impact of this work. We answer your questions below.
>
> ## More previous works in BNNs
>
> In the introduction section of our revised paper, we have incorporated the work you referenced, along with other pertinent studies. Specifically, we have expanded our discussion to include additional works on stochastic variational inference [2,3,4] and Monte Carlo methods [5,6,7], beyond those already cited in the current version.
>
>
> ## More experiments
>
> We extended our experiments to two distribution shift benchmarks including both Out-of-Distribution (OOD) detection tasks and robustness benchmarks. The results demonstrate that our SSVI consistently improves over the strongest baseline, mBNN, significantly across all sparsity levels, metrics, and tasks. The results are presented both in the subsequent sections and in the updated Appendix D.
>
> For OOD detection tasks, we computed the area under the receiver operating characteristic (AUROC) and the area under the precision-recall curve (AUPR) using our models, which were trained using SSVI on CIFAR-10/100 and tested on CIFAR-100/10 and SVHN, following the methodology of previous studies [1]. In this context, we employed the mBNN algorithm, the strongest previous work, as our baseline, ensuring equal sparsity for a fair and meaningful comparison.
>
> |  **Train set**  |  |           |   CIFAR-10   |       |      | |          |   CIFAR-100   |       |      |
> |:-----------:|:--------:|:---------:|:----:|:-----:|:----:|:---------:|:--------:|:----:|:-----:|:----:|
> |   Test set  | | CIFAR-100 |      |  SVHN |      |  | CIFAR-10 |      |  SVHN |      |
> |       Metric      |    Sparsity      |   AUROC   | AUPR | AUROC | AUPR |    Sparsity       |   AUROC  | AUPR | AUROC | AUPR |
> |     mBNN    |   96.4%  |    0.84   | 0.81 |  0.89 | 0.94 |   87.8%   |   0.76   | 0.72 |  0.78 | 0.88 |
> | SSVI (ours) |    95%   |    **0.86**   | **0.84** |  **0.93** | **0.96** |    **90%**    |   **0.80**   | **0.77** |  **0.85** | **0.91** |
>
>
> We proceed to assess our model, trained using SSVI on CIFAR-10/100, on the respective corrupted versions, CIFAR-10/100-C. Both datasets encompass 15 distinct corruptions, each classified into 5 corruption levels. The outcomes are presented in the following table, illustrating curves for the average accuracy, loss, and Expected Calibration Error (ECE) across all 15 different corruptions for each level.
>
> | Dataset       | Model              | Level 1 | Level 2 | Level 3 | Level 4 | Level 5 |
> |---------------|--------------------|---------|---------|---------|---------|---------|
> | CIFAR-10-C    | mBNN (ACC)         | 0.811   | 0.731   | 0.651   | 0.565   | 0.448   |
> |               | **SSVI (ACC)**     | **0.829**   | **0.754**   | **0.670**    | **0.585**   | **0.471**   |
> |               | mBNN (Loss)        | 0.721   | 1.075   | 1.469   | 1.931   | 2.659   |
> |               | **SSVI (Loss)**    | **0.668**   | **1.001**   | **1.384**   | **1.810**    | **2.408**   |
> |               | mBNN (ECE Loss)    | 0.066   | 0.092   | 0.110    | 0.129   | 0.151   |
> |               | **SSVI (ECE Loss)**| **0.052**   | **0.078**   | **0.099**   | **0.121**   | **0.145**   |
> |---------------|--------------------|---------|---------|---------|---------|---------|
> | CIFAR-100-C   | mBNN (ACC)         | 0.583   | 0.499   | 0.437   | 0.369   | 0.279   |
> |               | **SSVI (ACC)**     | **0.611**   | **0.522**   | **0.455**   | **0.388**   | **0.293**   |
> |               | mBNN (Loss)        | 2.062   | 2.69    | 3.320    | 4.045   | 5.156   |
> |               | **SSVI (Loss)**    | **1.725**   | **2.247**   | **2.821**   | **3.351**   | **4.090**    |
> |               | mBNN (ECE Loss)    | 0.015   | 0.023   | 0.026   | 0.033   | 0.039   |
> |               | **SSVI (ECE Loss)**| **0.012**   | **0.019**   | **0.024**   | **0.029**   | **0.036**   |
> |---------------|--------------------|---------|---------|---------|---------|---------|
>
> We agree that applying SSVI to more ML tasks, beyond standard models and applications, is an interesting question and we are happy to explore it further in our future work.

---

> ### Author Response · Authors · 2023-11-21
> **Response Part 2**
>
> ## Clarity
>
> Thank you for your thorough comments! We have made the suggested improvements to the paper. Please review the updated version. Here is a summary of responses to your questions:
>
> 1. The "20x compression, 20x FLOPs reduction" is benchmarked against the densely trained VI.
> 2. Our approach excels over VI-trained dense BNNs in terms of both accuracy and uncertainty metrics, such as the Expected Calibration Error (ECE).
> 3. The experimental section contains references and descriptions of the baselines.
> 4. The operator is element-wise matrix multiplication.
> 5. $B$ is batch size, $p,q$ are the input and output dimensions, $W$ is the weight matrix.
> 6. $l$ is the loss function received from the backward process.
> 7. Row 2 and 3 present the outcomes of our SSVI under two different sparsity levels, which are adjustable inputs in our SSVI.
> 8. We use equation (12) in our SSVI.
>
> ## References
>
> [1] Hippolyt Ritter, Martin Kukla, Cheng Zhang, and Yingzhen Li. Sparse uncertainty representation in deep learning with inducing weights. Advances in Neural Information Processing Systems, 34: 6515–6528, 2021.
>
> [2] Hoffman, Matthew D., et al. "Stochastic variational inference." Journal of Machine Learning Research (2013).
>
> [3] Dusenberry, Michael, et al. "Efficient and scalable bayesian neural nets with rank-1 factors." International conference on machine learning. PMLR, 2020.
>
> [4] Welling, Max, and Yee W. Teh. "Bayesian learning via stochastic gradient Langevin dynamics." Proceedings of the 28th international conference on machine learning (ICML-11). 2011.
>
> [5] Chen, Tianqi, Emily Fox, and Carlos Guestrin. "Stochastic gradient hamiltonian monte carlo." International conference on machine learning. PMLR, 2014.
>
> [6] Cobb, Adam D., and Brian Jalaian. "Scaling Hamiltonian Monte Carlo inference for Bayesian neural networks with symmetric splitting." Uncertainty in Artificial Intelligence. PMLR, 2021.
>
> [7] Wenzel, Florian, et al. "How good is the bayes posterior in deep neural networks really?." ICML 2020.

---

> > ### Comment · Reviewer_KZuX · 2023-11-22
> >
> > Thank you for your response. As I mentioned you didn't really have to response to me as I was quite happy with the initial version of the paper. Thank you though for addressing my comments. New results further convince in the usefulness of the proposed method.

---

### Official Review · Reviewer_DFxu · 2023-11-03

**Soundness:** 2 fair
**Presentation:** 2 fair
**Contribution:** 2 fair
**Rating:** 6
**Confidence:** 4

**Summary:**

In this paper, the authors present an approach to train sparse-BNNs using sparse subspace variational inference. The authors show that their approach gives computation gains during training as well as during test-time inference.

**Strengths:**

- The paper is well-motivated. Using BNNs in practice naively requires lot of compute which can hamper its adoption

- The experimental results provided by the paper are impressive. Having training time benefits is extremely useful.

- I also appreciate the fact that the authors have provided the implementation for their work.

**Weaknesses:**

- I think related relevant work is missing [1, 2]. Especially the work by Vadera et al.[1], looks at sparse SGHMC for BNNs, which could be easily extended to VI and when doing so would look similar to what the authors have proposed. The authors should highlight the difference between their work and existing work.

- The empirical results seem insufficient. The authors haven't mentioned the model architecture used in their experiments. Also, to really emphasize the usefulness of the proposed approach, it would be important to have empirical results on an expanded set of datasets + models + tasks (including UQ tasks such as OOD detection, misclassification detection, etc.)

- I think the presentation can be improved. The authors introduce complex notations, but after reading through section 3 twice, I believe that it can be greatly simplified. For e.g., where exactly is eq 2 reconciled in the algorithm? Based on my reading, the paper would have been fine to totally exclude eq 2.


References

[1] Vadera, M. P., Cobb, A. D., Jalaian, B., & Marlin, B. M. (2022). Impact of Parameter Sparsity on Stochastic Gradient MCMC Methods for Bayesian Deep Learning. arXiv preprint arXiv:2202.03770.

[2] Ghosh, S., Yao, J., & Doshi-Velez, F. (2018, July). Structured variational learning of Bayesian neural networks with horseshoe priors. In International Conference on Machine Learning (pp. 1744-1753). PMLR.

**Questions:**

- Is eq 2 being really used somewhere directly?

- What's the model architecture used in experiments?

- Apart from implementation, can the authors highlight the diff between their work and that of Vadera et al.?

- How would you extend this to other approximate Bayesian inference techniques, apart from mean-field VI?

- It'll be useful if the authors can demonstrate the quality of uncertainty metrics coming out of the BNNs in their approach on downstream tasks. See [3] for example.

References

[3] Vadera, M., Li, J., Cobb, A., Jalaian, B., Abdelzaher, T., & Marlin, B. (2022). URSABench: A system for comprehensive benchmarking of Bayesian deep neural network models and inference methods. Proceedings of Machine Learning and Systems, 4, 217-237.

---

> ### Author Response · Authors · 2023-11-21
> **Response Part 1**
>
> We are thankful for your valuable comments and inquiries which have contributed to a clearer understanding of key elements in our work. We believe the following explanations address all identified weaknesses and answer all the questions raised. If you find our response satisfactory, we kindly hope you will consider raising your score.
>
> ## Comparison with previous works [1, 2]
>
> We want to clarify that our work **fundamentally differs** from [1, 2]. Our approach involves developing a new framework for identifying the optimal sparse subspace, ensuring sparsity throughout the training process.  However, for [1], we quote what they mentioned at the beginning of their Section 3.1:
>
> *“ We emphasize that the contribution of this work is not to propose new methods for deriving sparse sub-networks, but rather to evaluate the impact of sparse sub-network structures on MCMC-based inference methods”*
>
> This indicates that [1] focuses on applying MCMC to pre-existing sparse neural networks to assess their impact on MCMC inference, which significantly differs from our work. While it is possible to modify their approach for VI, such as implementing VI on these pre-existing networks, our work diverges entirely from this route in two points:
> Our work aims to develop a novel algorithm for choosing sub-networks instead of just using existing methods.
> In contrast to the two-step method of first developing a sparse network and then conducting MCMC in [1], our approach integrates the selection of sub-networks directly into the VI training process, creating a more efficient, single-phase procedure.
>
> Similar to several studies we discussed in the related works section that differ from our SSVI, [2] employs a one-shot pruning method following dense training, leading to significantly increased training costs compared with our SSVI. This one-shot approach tends to result in sub-optimal sub-networks, unlike our method which iteratively removes and adds elements to pinpoint a better sub-network. Furthermore, [2] depends on complexly designed priors to induce sparsity in the model, which can result in unpredictable levels of final sparsity. In contrast, our approach allows for precise control over the final sparsity level right from the start of the training process.
> We have included [1,2] in the related works in the revision.
>
> ## The authors haven't mentioned the model architecture used in their experiments.
>
> We have mentioned the model architecture used in experiments, ResNet-18, in the first line of Section 4.1 on page 7. This model architecture is standard and has been widely used in the sparse Bayesian Neural Network (BNN) literature such as mBNN (Kong et al., 2023) and AEB (Deng et al., 2019). In fact, many papers on sparse BNNs (including Kong et al., 2023 and Deng et al., 2019) used ResNet as the **only** model architecture for large-scale Bayesian deep learning experiments. Therefore, we also chose ResNet to ensure a fair comparison.

---

> ### Author Response · Authors · 2023-11-21
> **Response Part 2**
>
> ## It would be important to have empirical results on an expanded set of datasets + models + tasks
>
> We extended our experiments to two distribution shift benchmarks including both Out-of-Distribution (OOD) detection tasks and robustness benchmarks. The results demonstrate that our SSVI consistently improves over the strongest baseline, mBNN, significantly across all sparsity levels, metrics, and tasks. The results are presented both in the subsequent sections and in the updated Appendix D.
>
> For OOD detection tasks, we computed the area under the receiver operating characteristic (AUROC) and the area under the precision-recall curve (AUPR) using our models, which were trained using SSVI on CIFAR-10/100 and tested on CIFAR-100/10 and SVHN, following the methodology of previous studies [1]. In this context, we employed the mBNN algorithm, the strongest previous work, as our baseline, ensuring equal sparsity for a fair and meaningful comparison.
>
> |  **Train set**  |  |           |   CIFAR-10   |       |      | |          |   CIFAR-100   |       |      |
> |:-----------:|:--------:|:---------:|:----:|:-----:|:----:|:---------:|:--------:|:----:|:-----:|:----:|
> |   Test set  | | CIFAR-100 |      |  SVHN |      |  | CIFAR-10 |      |  SVHN |      |
> |       Metric      |    Sparsity      |   AUROC   | AUPR | AUROC | AUPR |    Sparsity       |   AUROC  | AUPR | AUROC | AUPR |
> |     mBNN    |   96.4%  |    0.84   | 0.81 |  0.89 | 0.94 |   87.8%   |   0.76   | 0.72 |  0.78 | 0.88 |
> | SSVI (ours) |    95%   |    **0.86**   | **0.84** |  **0.93** | **0.96** |    **90%**    |   **0.80**   | **0.77** |  **0.85** | **0.91** |
>
>
> We proceed to assess our model, trained using SSVI on CIFAR-10/100, on the respective corrupted versions, CIFAR-10/100-C. Both datasets encompass 15 distinct corruptions, each classified into 5 corruption levels. The outcomes are presented in the following table, illustrating curves for the average accuracy, loss, and Expected Calibration Error (ECE) across all 15 different corruptions for each level.
>
> | Dataset       | Model              | Level 1 | Level 2 | Level 3 | Level 4 | Level 5 |
> |---------------|--------------------|---------|---------|---------|---------|---------|
> | CIFAR-10-C    | mBNN (ACC)         | 0.811   | 0.731   | 0.651   | 0.565   | 0.448   |
> |               | **SSVI (ACC)**     | **0.829**   | **0.754**   | **0.670**    | **0.585**   | **0.471**   |
> |               | mBNN (Loss)        | 0.721   | 1.075   | 1.469   | 1.931   | 2.659   |
> |               | **SSVI (Loss)**    | **0.668**   | **1.001**   | **1.384**   | **1.810**    | **2.408**   |
> |               | mBNN (ECE Loss)    | 0.066   | 0.092   | 0.110    | 0.129   | 0.151   |
> |               | **SSVI (ECE Loss)**| **0.052**   | **0.078**   | **0.099**   | **0.121**   | **0.145**   |
> |---------------|--------------------|---------|---------|---------|---------|---------|
> | CIFAR-100-C   | mBNN (ACC)         | 0.583   | 0.499   | 0.437   | 0.369   | 0.279   |
> |               | **SSVI (ACC)**     | **0.611**   | **0.522**   | **0.455**   | **0.388**   | **0.293**   |
> |               | mBNN (Loss)        | 2.062   | 2.69    | 3.320    | 4.045   | 5.156   |
> |               | **SSVI (Loss)**    | **1.725**   | **2.247**   | **2.821**   | **3.351**   | **4.090**    |
> |               | mBNN (ECE Loss)    | 0.015   | 0.023   | 0.026   | 0.033   | 0.039   |
> |               | **SSVI (ECE Loss)**| **0.012**   | **0.019**   | **0.024**   | **0.029**   | **0.036**   |
> |---------------|--------------------|---------|---------|---------|---------|---------|
>
>
> ## Notations and Equation 2
>
> Our intention behind introducing all notations is to provide a clearer illustration of both our intuition and algorithm. Regarding Equation 2, we want to emphasize that it serves as the objective function (loss function) for our Sparse Subspace Variational Inference (SSVI). This is similar to Equation 1, which is the objective function for standard Variational Inference (VI). Equation 2 is a slight modification of Equation 1, representing our goal to constrain the variational space to a sparse subspace. The proposed algorithm SSVI is then developed to solve the optimization problem stated in Equation 2.
>
> Moreover, Equation 2 serves as the general form for sparse subspace VI. If we choose the common Gaussian as the variational distribution, it will result in a special case as shown in Equation 3. It is worth noting that other distribution families are possible but they may introduce unnecessary complexity and lead to alternative forms based on Equation 2.
>
> In summary, Equation 2 is important to our framework, providing the foundation for the problem this paper tackles and guiding subsequent developments in our approach. We will revise this point in the final version to improve the clarity of the paper.

---

> > ### Comment · Reviewer_DFxu · 2023-11-22
> > **Response to the rebuttal (Part 2)**
> >
> > Many thanks for running the additional experiments. I now realise that it's important to note why mBNN is the only/most useful baseline here.
> >
> > Point noted on equation 2. Is this already updated in the paper? If yes, could you please point me to it?

---

> > > ### Author Response · Authors · 2023-11-22
> > > **Author Response Round 2 (Part 2)**
> > >
> > > Thanks a lot for your acknowledgment of our additional experiments!
> > >
> > > We just simplified equation (2) to better align it with equation (3). Please review our updated paper.

---

> ### Author Response · Authors · 2023-11-21
> **Response Part 3**
>
> ## Extention to other approximate Bayesian inference techniques
>
> Bayesian neural networks based on other approximate Bayesian inference techniques, such as non-mean-field VI and MCMC, often involve distinct assumptions and approximations, such as those outlined in [4,5], which are beyond the scope of our current work as a conference submission. However, we appreciate the significance of this direction and plan to explore it in our future work.
>
> ## Benefits of uncertainty metric in downstream tasks
>
> We believe that the additional experiments on out-of-distribution (OOD) detection and distribution shift tasks, as outlined above and detailed in Appendix D.1, effectively address this question. The experiments we have are similar to those in [3]. Our SSVI surpasses all baseline models in terms of ECE for standard tasks, AUROC and AUPR for OOD tasks. This improvement in uncertainty metrics contributes to the enhanced accuracy on corrupted datasets, as evidenced by our results with CIFAR-10/100-C.
>
> ## References
>
> [1] Hippolyt Ritter, Martin Kukla, Cheng Zhang, and Yingzhen Li. Sparse uncertainty representation in deep learning with inducing weights. Advances in Neural Information Processing Systems, 34: 6515–6528, 2021.
>
> [2] Ghosh, S., Yao, J., & Doshi-Velez, F. (2018, July). Structured variational learning of Bayesian neural networks with horseshoe priors. In International Conference on Machine Learning (pp. 1744-1753). PMLR.
>
> [3] Vadera, M., Li, J., Cobb, A., Jalaian, B., Abdelzaher, T., & Marlin, B. (2022). URSABench: A system for comprehensive benchmarking of Bayesian deep neural network models and inference methods. Proceedings of Machine Learning and Systems, 4, 217-237.
>
> [4] Nguyen, Son, et al. "Structured dropout variational inference for Bayesian neural networks." Advances in Neural Information Processing Systems 34 (2021): 15188-15202.
>
> [5] Zhang et al. Cyclical Stochastic Gradient MCMC for Bayesian Deep Learning. ICLR 2020.

---

> > ### Comment · Reviewer_DFxu · 2023-11-22
> > **Response to the rebuttal (Part 3)**
> >
> > Please add the extension to other BNN algorithms as a part of the future work (and it would be great to discuss some directions).
> >
> > I've already responded to the new results in my previous response - so I am not going to add it again here.

---

> > > ### Author Response · Authors · 2023-11-22
> > > **Author Response Round 2 (Part 3)**
> > >
> > > Thanks again for your comments! We will include this important and promising direction in our future works!

---

> ### Comment · Reviewer_DFxu · 2023-11-22
> **Response to the rebuttal (Part 1)**
>
> I thank the authors for engaging in the rebuttal.  I'll respond to their rebuttal part by part.
>
> Furthermore, I believe the above responses are fair - however, I don't quite agree with the characterization of on work by Ghosh et al., 2019 [1]. The crux of the work is to use sparsity inducing priors for VI - the sparsity isn't induced by dense training. Could the authors elaborate more on this?
>
> Finally, while I agree that the work of Vadera et al. doesn't really develop novel techniques for finding optimal sparse substructure, I still believe that it's an important baseline to compare against.
>
>
> References
>
> [1] Ghosh, S., Yao, J., & Doshi-Velez, F. (2018, July). Structured variational learning of Bayesian neural networks with horseshoe priors. In International Conference on Machine Learning (pp. 1744-1753). PMLR.

---

> ### Author Response · Authors · 2023-11-22
> **Author Response Round 2 (Part 1)**
>
> Thanks again for your careful review and acknowledgment of our response and detailed reply!
>
> ## The characterization of Ghosh et al., 2019 [1]
>
> We want to clarify that we actually both agree on the point that  the sparsity is induced by priors for VI, as highlighted in both our first-round rebuttal: *“[1] depends on complexly designed priors to induce sparsity in the model, ”* and also in our related works section in the original paper *“many studies have employed sparse-promoting priors to obtain sparse BNNs.”* and *“Louizos et al. (2017), Ghosh et al. (2019) and Bai et al. (2020) further explore the use of half-Cauchy, Horseshoe and Gaussian-Dirac spike-and slab priors respectively.”*
>
> We are not suggesting that dense training leads to sparsity. Our reference to "dense training" is to highlight our SSVI's superiority and distinction from previous works, including Ghosh et al., 2019 [1]. These studies induce sparsity through the use of specific priors. However, as noted in Section 4.3 of Ghosh et al., 2019 [1], *“The Horseshoe and its regularized variant provide strong shrinkage towards zero for small $w_{kl}$. However, the shrunk weights, although tiny, are never actually zero. ”*, this means their training is still dense so they still have a high training cost. In contrast, our SSVI approach achieves sparsity right from the start, leading to reductions in both training and inference costs.
>
> ## About Vadera et al.
>
> We have already added this paper Vadera et al. to our introduction section. We also agree that it’s an interesting direction to extend the method from MCMC to VI as outlined in Vadera et al., and evaluate the sparse structure of our SSVI in comparison to prior works using approaches from Vadera et al. However, this is beyond the scope of our current work. We will investigate this systematically in our future work.

---

> ### Author Response · Authors · 2023-11-22
> **Thank you for your time and valuable input!**
>
> Thank you once more for acknowledging our rebuttal and the additional experiments we conducted. In your latest response, you raised four points:
>
> 1. The characterization of Ghosh et al., 2019 [1] – we have clarified this and confirmed that our views are in sync with yours, indicating no divergence between us on this aspect.
> 2. Simplification of Equation 2 – we have already made these simplifications in our revised paper.
> 3. Comparison with Vadera et al. – we acknowledge this valuable direction and plan to include it in our future research.
> 4. Extension to other BNNs – this direction too will be addressed in our upcoming work.
>
> We are confident that we have adequately addressed these comments, as well as those from your first-round review. If you find our responses to be satisfactory, we would appreciate it if you would consider raising your score for our submission. Thank you for your time and valuable input!

---

> > ### Comment · Reviewer_DFxu · 2023-11-22
> > **Thanks for your response**
> >
> > I thank the authors for engaging in the discussion. I will be raising my score. Good luck!

---

### Author Response · Authors · 2023-11-21
**Thank you to all reviewers for your valuable and constructive feedback!**

We are grateful to all the reviewers for your thorough examination and valuable feedback on our work. We are confident that we address all raised issues and invite you to review our responses. The majority of the modifications are presented here in several parts. Additionally, we have incorporated updates and figures in our paper and supplementary materials in line with your comments and suggestions. Specifically, we:

* Introduced a new section, Appendix D, which contains further experiments. Appendix D.1 presents experiments on distribution shift tasks including both Out-of-Distribution (OOD) tasks and robustness benchmarks. In Appendix D.2, we conduct additional ablation studies highlighting the significance of our proposed removal criteria.

* Added more references in the introduction section.
* Refined various statements throughout the paper to enhance clarity.

---

### Meta-Review · Area_Chair_Q8rs · 2023-12-06

**Metareview:**

This paper proposes a novel framework to introduce sparsity into BNNs using certain weight distribution statistics. It shows that this can drastically reduce the model size and runtime during training with only minimal cost in performance. After an active discussion between authors and reviewers, all reviewers lean towards acceptance. While the reviewers praised the importance of the problem, the strong empirical results, and the theoretical foundations, they were critical of the treatment of related work, the narrow set of experiments, the lack of ablation studies, and the clarity of the writing. However, most (if not all) of these issues have been addressed in the extensive rebuttal. It therefore seems warranted to accept the paper, with the understanding that the authors will continue their efforts to fully address the reviewer feedback in the camera-ready version.

**Justification For Why Not Higher Score:**

it is unclear that this paper would be interesting for a wide enough audience to warrant a spotlight

**Justification For Why Not Lower Score:**

all reviewers agree to accept

---

### Decision · Program_Chairs · 2024-01-16

Accept (poster)